# DORIS-MAE:
# Scientific Document Retrieval using Multi-level Aspect-based Queries

**Jianyou Wang** *      **Kaicheng Wang** *      **Xiaoyue Wang**      **Prudhviraj Naidu**

**Leon Bergen** †                **Ramamohan Paturi** †

Laboratory for Emerging Intelligence
University of California, San Diego
La Jolla, CA 92093
{jiw101, kaw036, xiw027, prnaidu, lbergen, rpaturi} @ ucsd.edu

## Abstract

In scientific research, the ability to effectively retrieve relevant documents based on complex, multifaceted queries is critical. Existing evaluation datasets for this task are limited, primarily due to the high cost and effort required to annotate resources that effectively represent complex queries. To address this, we propose a novel task, **S**cientific **DO**cument **R**etrieval using **M**ulti-level **A**spect-based qu**E**ries (DORIS-MAE), which is designed to handle the complex nature of user queries in scientific research. We developed a benchmark dataset within the field of computer science, consisting of 100 human-authored complex query cases. For each complex query, we assembled a collection of 100 relevant documents and produced annotated relevance scores for ranking them. Recognizing the significant labor of expert annotation, we also introduce Anno-GPT, a scalable framework for validating the performance of Large Language Models (LLMs) on expert-level dataset annotation tasks. LLM annotation of the DORIS-MAE dataset resulted in a 500x reduction in cost, without compromising quality. Furthermore, due to the multi-tiered structure of these complex queries, the DORIS-MAE dataset can be extended to over 4,000 sub-query test cases without requiring additional annotation. We evaluated 17 recent retrieval methods on DORIS-MAE, observing notable performance drops compared to traditional datasets. This highlights the need for better approaches to handle complex, multifaceted queries in scientific research. Our dataset and codebase are available at https://github.com/Real-Doris-Mae/Doris-Mae-Dataset.

## 1   Introduction

Scientists often have complex questions that require thorough exploration within various parts of their field (Figure 1). Finding relevant scientific literature, one of many challenges in this process, can be especially difficult when dealing with multi-faceted queries. These queries typically encompass numerous interconnected topics and require an information retrieval (IR) system capable of recognizing and responding to this level of complexity.

---

*Equal Contribution
†Equal Contribution

37th Conference on Neural Information Processing Systems (NeurIPS 2023) Track on Datasets and Benchmarks.

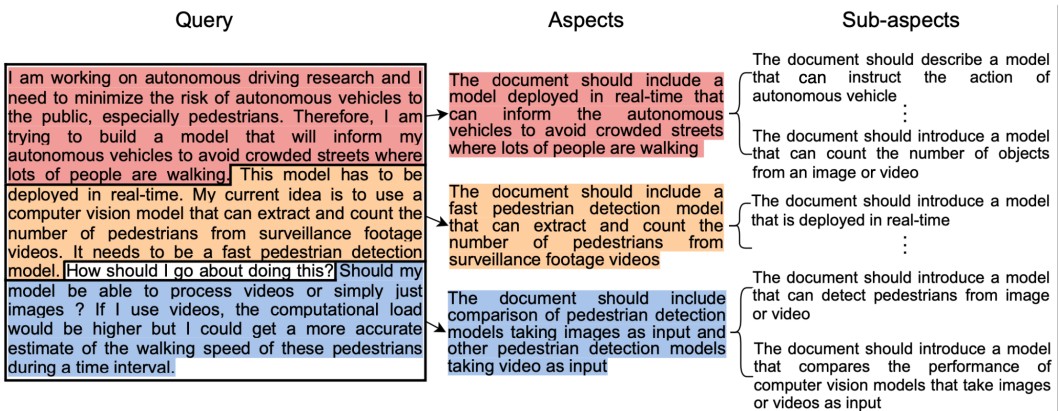

Figure 1: Example from the DORIS-MAE dataset. Each query is broken down into aspects and sub-aspects. Aspects are semantically distinct components of the query, and sub-aspects are minimal requirements that can be extracted from the aspects.

Information retrieval, especially query-based document retrieval [78, 3, 34], is integral to many applications, from search engines [9, 13, 29] and content recommendations [58, 31, 41], to open-domain question answering (QA) [82, 33, 15]. A persistent challenge, however, is the low accuracy in processing complex and multi-intent user queries. Despite advanced search engines using semantic understanding and user behavior data in addition to keyword matching [13], these systems still fall short when dealing with complex, multi-intent queries. Neural Information Retrieval (NIR) models [63, 85, 75, 23, 66, 85, 47] are primarily trained on relatively simple queries [57, 17, 64, 61, 87]. Moreover, MacAvaney et al. [49] creates a variety of diagnostic probes, revealing NIR models' instability when processing out-of-distribution textual inputs, potentially because they do not comprehend the deeper semantics of text. These limitations can lead to inadequate performance with more complex queries. While open-domain QA models like ChatGPT [11] have shown significant capability, they frequently produce incorrect or even fabricated responses [4, 46, 50, 39, 89], and are prohibitively expensive if directly applied to an entire corpus of scientific documents.

To address these challenges, we introduce a novel task, **S**cientific **Do**cument **R**etrieval for **M**ulti-level **A**spect-based qu**E**ries (DORIS-MAE). DORIS-MAE extends query-based and example-based IR paradigms [20, 44], aiming to give users more control in formulating queries using natural language. In order to advance research in this area, we present the DORIS-MAE dataset, comprising 100 unique complex queries in the computer science domain, paired with ranked pools of relevant CS article abstracts. Each query is organized into a hierarchical structure of aspects/sub-aspects, which aids annotation.

A distinguishing feature of the DORIS-MAE dataset is its aspect-based hierarchical structure shown in Figure 1. This feature aids automation of the annotation process, expands the test case volume without additional annotation, and opens up exploration into the usefulness of aspect information for retrieval methods.

Furthermore, we introduce Anno-GPT, a pipeline for validating Large Language Model (LLM) annotations in a statistically sound manner. Our tests reveal that LLM annotations achieve quality comparable to those produced by human annotators, but with considerable savings in both time and cost. Additionally, the design of our pipeline lends itself to easy adaptation for different domains.

In our experiments, we evaluated 17 IR/NIR (Information Retrieval/Neural Information Retrieval) methods using the DORIS-MAE dataset. The methods have worse performance on DORIS-MAE compared to traditional document retrieval datasets, highlighting the complexity of DORIS-MAE, and the need for more sophisticated retrieval methods.

Our main contributions are three-fold. First, by formulating the **DORIS-MAE** task we are shedding a new light on tackling complex, multi-faceted queries during scientific research. Second, we propose **Anno-GPT**, a procedure for rigorously evaluating the ability of LLMs to replace human experts for challenging annotation tasks. Third, we demonstrate the value of breaking complex queries down

to an **Multi**-level **Aspect**-based hierarchical structure, for both annotation accuracy and potential improvements in retrieval methods.

## 2    Related Work

A range of methods have been developed for document retrieval and re-ranking. Classic retrieval methods like TF-IDF [67] and BM25 [72] utilize keyword matching between queries and documents, and fall short when the necessary key phrases are not known to users. To address this, researchers have applied deep learning techniques to develop NIR models, including RocketQA-v2 [63], ColBERT-v2 [66], SimLM [75], and SPLADE-v2 [23]. These models generate latent vector representations for queries and documents, which are effective for many retrieval tasks, but may not be able to handle textual inputs that are outside of their training distributions [49].

Other models such as SPECTER [18] and aspect-based ASPIRE [54] focus on calculating document-level similarity. These models are designed to retrieve given an existing paper as input, which makes them less suited for open-ended queries. Other retrieval models, such as text-embedding models (ada-002 by OpenAI [28] and E5-large-v2 [74]) and sentence-embedding model Sentence-BERT [62], all face similar limitations as NIR methods in that they struggle to simultaneously represent multiple aspects of a query. Despite these challenges, they tend to perform better on our dataset than specialized NIR models for efficient dense passage retrieval, indicating their potential for complex tasks like DORIS-MAE.

Large-scale traditional IR datasets [57, 86, 40, 66, 69] primarily contain simple, web-based queries. Models performing well on these datasets often struggle on DORIS-MAE. In contrast, more specialized IR datasets [56, 14, 53] require human expert annotation, and consequently are more limited in their scale. Mysore et al. [53] and Chan et al. [14] introduce the concept of "aspect" in document retrieval tasks. These datasets provide pre-defined categories for aspects such as "background" or "method". DORIS-MAE extends this work by allowing for open-ended aspects based on the context of the queries. In Narrative-Driven Recommendation (NDR) research [6, 2, 55], user queries are descriptions that capture a range of users' needs. These queries are related to everyday tasks, such as finding restaurants and entertainment.

Progress in LLMs [11] and prompt-engineering [24, 45, 59, 79, 65] has made it feasible to leverage LLMs such as ChatGPT for annotating NLP tasks [90, 21], even outperforming crowd-workers in some cases [60, 70, 27]. However, these tasks do not necessitate domain-specific knowledge, and the annotations produced may not measure up to expert annotations. Faggioli et al. [22], MacAvaney and Soldaini [48] explored the notion of using LLMs to assist human in relevance judgements, and evaluated on TREC-DL datasets [19], which have single-faceted queries. Our work seeks to extend these efforts to new annotation tasks requiring domain expertise, while introducing a separate pipeline stage and a hypothesis testing stage.

## 3    Dataset Description

The DORIS-MAE task uses a dataset of 100 complex, human-written queries, each containing between 95 to 226 words. Each complex query is broken down into a hierarchy of aspects and sub-aspects, with aspects representing significant semantic components of the original query, typically a sentence or a few sentences. Sub-aspects further decompose an aspect into simpler, verifiable requirements. Both aspects and sub-aspects are generally one sentence long, though their semantic complexity varies. A complex query can have up to 9 aspects, and each aspect can further contain up to 6 sub-aspects. Figure 1 shows an example.

For each complex query $Q$, we created a pool $\mathcal{P}$ of approximately 100 potentially relevant scientific abstracts to evaluate the re-ranking performance of various retrieval/reranking methods. Within the context of $Q$ and its $\mathcal{P}$, the collection of aspects and sub-aspects is together denoted as (with slight abuse of notation) $Q := \{a_i\}$. For any aspect or sub-aspect $a_i \in Q$, and any paper abstract $p_j \in \mathcal{P}$, they form a question pair $(a_i, p_j)$. We generated a total of 165,144 question pairs from the 100 queries, together denoted as $\mathcal{D} := \{(a_i, p_j)\}$.

To compute the relevance of a paper $p_j$ for a query $Q$, we first break the query down into its aspects. We then compute the relevance score $S(a_i, p_j)$, which measures the relevance of paper $p_j$ for aspect

$a_i$. The relevance score $S(p_j|Q)$ for the query $Q$ is the sum of its aspect relevance scores. An average score $\frac{S(p_j|Q)}{|Q|} \geq 1$ indicates $p_j$ is a relevant abstract for $Q$.

$$S(p_j|Q) := \sum_{a_i \in Q} S(a_i, p_j) \tag{1}$$

Thus, for each complex query and candidate pool, we provide a complete ranking for the list of abstracts. Additionally, we can choose a combination of aspects within a complex query $Q$ and concatenate the corresponding sentences to form a sub-query $q \subset Q$, and can calculate the relevance score between a paper abstract $p_j$ and a sub-query $q$.

$$S(p_j|q) := \sum_{a_i \in q} S(a_i, p_j) \tag{2}$$

Hence, for any sub-query $q \subset Q$ and the candidate pool $\mathcal{P}$ of $Q$, we can also provide a complete ground-truth ranking order for $\mathcal{P}$. This allows the dataset to extend to over 4000 sub-query test cases at no additional annotation cost.

Candidate pool abstracts are taken from a database of 360,000 computer science papers between 2011-2021 sourced from arXiv[3]. We complemented each arXiv paper with its corresponding citation information by cross-matching it on Semantic Scholar [37].

## 3.1 Query Formation

The DORIS-MAE task aims to mirror real-world scenarios where a researcher has an incomplete concept for a research project and needs to explore the breadth of existing research to establish a solid starting point. The 100 complex queries in our dataset simulate this scenario.

Each query is based on one or more existing research papers. We randomly selected 140 papers from AI, NLP, ML, and CV categories on arXiv. We examined each selected paper's motivation, background, related work, methodology, and experimental results. Using this information, we reverse-engineered a complex query designed to reflect the early thought process of the paper's authors. Overall, DORIS-MAE contains 80 queries derived from single paper abstracts, with 20 queries from each of AI, NLP, ML and CV. We created an additional 20 composite queries, each integrating ideas from 2-3 abstracts. These composite queries are designed to simulate more interdisciplinary and unexplored research ideas than the other 80 queries.

## 3.2 Decomposing Queries to Aspects

In this section, we discuss our process of decomposing queries into a list of aspects and further breaking down aspects into sub-aspects. The guiding principles for determining aspects are as follows:

(i) Each aspect must correspond to a prominent and semantically meaningful component in the query (refer to Figure 1 for this correspondence).

(ii) Each aspect should contain sufficient context to make sense independently, eliminating potential ambiguities.

(iii) Each aspect must be semantically distinct and unrelated to others, ensuring their contents are disjoint and do not overlap.

Some aspects may fall into broader facets such as *background*, *method*, or *result* as defined in [53]. Regarding decomposing aspects to sub-aspects, our criteria are:

(i) Each sub-aspect should not contain more information than its parent aspect.

(ii) Each sub-aspect should represent a semantic segment of its parent aspect

(iii) Different sub-aspects may overlap semantically but each should pose a distinct question.

(iv) Each sub-aspect should be as simple as possible.

---

[3]https://arxiv.org/

Given the inherent difficulty in recalling the full details of a query while ranking a large candidate pool, the aspect-based hierarchical structure of our complex queries provides a systematic, efficient, and interpretable approach for annotation. This leads to a more precise ranking of the candidate pool. As each aspect corresponds to a specific part of the original query $Q$, combining several aspects is equivalent to concatenating their corresponding parts in the query, forming semantically coherent sub-queries $q \subset Q$.

### 3.3 Candidate Pooling

To create a candidate pool $\mathcal{P}$ of approximately 100 paper abstracts for each query $Q$, we used a variety of IR and NIR methods, similar to the approach in [53]. More specifically, we utilized popular IR search algorithms TF-IDF [67] and BM25 [72] at different granularities (i.e., sentence and paragraph-level for each query) to retrieve around 80 paper abstracts. We then employed OpenAI's text embedding model, ada-002 [28], to extract 20 more paper abstracts. We also use citation signals from semantic scholar [37] to included any papers that directly cite or are cited by any of the original papers used to create that specific query. Lastly, to prevent bias towards lexicon-matching retrieval methods, we excluded from $\mathcal{P}$ any papers that authors referenced during query $Q$'s formulation. See Appendix C.6 for a sensitivity analysis of our candidate pool construction procedure.

## 4 Anno-GPT Framework

We propose Anno-GPT, a framework for developing a statistically sound annotation pipeline. We use chatgpt-3.5-turbo-0301 to annotate all 165,144 question pairs $\mathcal{D} := \{(a_i, p_j)\}$ where $a_i$ is an aspect/sub-aspect of a query, and $p_j$ is an abstract in the query's candidate pool. This strategy minimizes human annotation efforts. The key to this approach lies in breaking down complex queries into simpler aspects and sub-aspects, ensuring an objective and manageable annotation task. Without this structured approach, we found that evaluating an abstract's relevance to a complex query was challenging, due to variability in how partial relevance was assessed. However, the scenario changes significantly when dealing with question pairs $(a_i, p_j)$, which only look at a single aspect or sub-aspect of the query. In such cases, assigning a coarse-scale relevance score between 0-2 becomes feasible, maintaining a reasonable degree of objectivity.

The performance of the annotation pipeline may be influenced by several factors: the procedure for breaking down queries into aspects, the criteria used for scoring query relevance, and the LLM prompt selection [24, 45, 59, 79, 65]. In order to avoid overfitting in this pipeline, our methodology comprises two distinct development and testing stages. The development stage involves optimizing all stages of the pipeline, and using feedback and observed outcomes to iteratively refine this strategy. The testing stage uses a prespecified hypothesis test. The null hypothesis is that there is no significant difference between the agreement levels of ChatGPT and humans, and those among humans themselves. After the pipeline has been optimized, we evaluate it on a test set. We compute the difference between ChatGPT-human agreement and human-human agreement. If this difference is sufficiently close to 0, with a small confidence interval, we can use the optimized prompt $\pi$ to annotate the full dataset.

Below we give a summary of the Anno-GPT framework:

    i Construct aspect-document question pairs, $\mathcal{D} := \{(a_i, p_j)\}$.
    ii Select development set $S_{\text{dev}} \subset \mathcal{D}$, and use human annotators to score $S_{\text{dev}}$.
    iii Select test set $S_{\text{test}} \subset \mathcal{D} \setminus S_{\text{dev}}$ based on desired power, and use human annotators to score $S_{\text{test}}$.
    iv Optimize prompting strategy $\pi$ and other hyperparameters on $S_{\text{dev}}$. Fix $\pi$.
    v Evaluate fixed $\pi$ on $S_{\text{test}}$.
    vi If satisfactory performance is achieved on $S_{\text{test}}$, proceed to use $\pi$ to annotate the entire dataset $\mathcal{D}$.
    vii Otherwise, repeat steps iii, iv, v for $S_{\text{dev}} \leftarrow S_{\text{dev}} \cup S_{\text{test}}$ and new $S_{\text{test}}$.

It is important to note that automated annotation for DORIS-MAE only used a single development/testing cycle, and therefore did not go into step vii. Therefore, there was no risk of inflated estimates of annotation accuracy due to multiple comparisons [30, 83, 1]. If multiple cycles are necessary, then the new $S_{\text{test}}$ must be sufficiently large to avoid these problems.

## 4.1 Annotation Guidelines

Our team of annotators consists of three graduate students in computer science, all with at least two years of research experience in NLP, CV, ML, and AI. Annotators, both human and ChatGPT, are asked to score each question pair using a 3-point grading scale (0-2):

- **Level 0:** The abstract is unrelated or does not provide any help to the key components of the aspect or sub-aspect.
- **Level 1:** The abstract answers some (or all) key components (either explicitly or within one natural inferential step), but at least one key component is not answered explicitly.
- **Level 2:** The abstract directly answers all the key components of the aspect or sub-aspect.

We decided to include both direct and indirect coverage under Level 1, acknowledging that distinguishing between these cases could be challenging and potentially subjective. The detailed guidelines for human annotation can be found in Appendix B.

## 4.2 Optimization of Annotation Pipeline

For the development stage, two annotators independently annotated a randomly selected set $S_{\text{dev}}$ of 90 question pairs from the complete set $\mathcal{D}$ of 165,144 pairs. This annotated development set then served as the basis for refining the prompting strategies for ChatGPT.

The quality of annotations was evaluated using three metrics: macro F1 score, exact accuracy (agreement), and Spearman's rank correlation coefficient (Spearman's $\rho$). These metrics measure the agreement level between annotators, and have been used successfully in similar tasks [53]. After satisfactory agreement levels were achieved between ChatGPT and human annotations on the development set, we transitioned to the hypothesis testing stage, with all three annotators involved. In this stage, we employed the fixed finalized prompting strategy, ensuring no overfitting or leakage from the test set.

Our prompting strategy development involved experimenting with recent methodologies such as few-shot in-context learning (ICL) [16, 11, 80, 51, 91] and chain of thought (CoT) [38, 81, 77, 32]. We found that the CoT approach offered the most robust and optimal results for the task of annotating question pairs $(a_i, d_j) \in \mathcal{D}$. A comprehensive description of the prompt engineering process can be found in Appendix B.

## 4.3 Annotation Evaluations

Hypothesis testing was conducted using a sample of 250 question pairs $S_{\text{test}}$ from $\mathcal{D}$, distinct from the development set. The selected pairs were independently annotated by three human annotators. We used bootstrapping to estimate the 95% confidence intervals for the macro F1 score, accuracy (agreement), and Spearman's $\rho$. Though the sampling temperature is fixed at zero, recognizing small randomness introduced by GPU non-determinism [52], the ChatGPT annotations are run twice and the pairwise comparisons with human annotators are averaged across these runs.

In addition to these prespecified analyses, we conducted post-hoc analyses using an adjudication procedure to create a more stable set of human annotations [53]. We use majority voting [8] between the three human annotators to decide the adjudicated annotation.

The results presented in Table 1 show that the rate of ChatGPT-human agreement is within range of human-human agreement. Specifically, ChatGPT's performance is comparable to that of human annotators as measured by F1 and exact agreement (accuracy). The average agreement level for ChatGPT is numerically lower than average human agreement level as measured by Spearman's $\rho$. We note that the lowest Spearman's $\rho$ among two humans is $46.51\%$, which is comparable to the average ChatGPT/human's $\rho = 46.61\%$, suggesting ChatGPT's performance is still within the range of human-level agreement. All $p$-values are larger than the $\alpha = 0.05$ criterion.

In a post-hoc analysis, we found that comparing ChatGPT to adjudicated human annotations numerically increased the rate of agreement. This provides qualitative evidence for ChatGPT's performance relative to that of human experts. We further analyzed instances where ChatGPT's annotations diverged from those of humans. Interestingly, the nature of these discrepancies was similar to those found between humans, with differences largely revolving around the interpretation of key components in aspect/sub-aspects. For example, ChatGPT occasionally differed from human annotators in

determining the importance of a given component. Detailed examples of ChatGPT's reasoning and a comprehensive error analysis can be found in Appendix B.

Table 1: Annotation agreement between humans and ChatGPT. H is human, G is ChatGPT, A is Adjudication, CI is 95% confidence interval. $p$-values correspond to the null hypothesis that there is no difference between avg. H&H and avg. H&G. Higher $p$-values indicate less evidence of a difference between ChatGPT and humans.

| Metrics | G&A | avg. H&H | avg. G&H | H&H CI | G&H CI | $p$-value |
|---|---|---|---|---|---|---|
| F1-score (macro) | 64.17 | 58.33 | 57.46 | (52.33, 63.46) | (50.93, 62.79) | 0.74 |
| Accuracy | 67.40 | 64.13 | 62.07 | (59.73, 68.80) | (57.67, 66.13) | 0.41 |
| Spearman's $\rho$ | 52.63 | 54.31 | 46.61 | (46.87, 61.56) | (38.67, 54.41) | 0.07 |

### 4.4 Scalability of Annotations

The hypothesis testing results support the use of ChatGPT for annotation. At deployment, the pipeline annotated all 165,144 aspect-paper pairs within a span of 24 hours, at a cost under $150. By contrast, human experts typically require approximately 4 minutes per question pair, resulting in an estimated 11,146 hours to annotate the entire dataset. The deployment resulted in a time reduction by a factor of 500, and a cost reduction by a factor of 1,000, without sacrificing annotation quality. Upon completion of the annotation process, we utilized Equations 1 and 2 to compile the results and compute the final rankings for both full-query and sub-query test cases. Anno-GPT could potentially utilize any high-performance LLM to replace ChatGPT and can be adapted for other expert-level tasks, given the availability of a small set of domain expert annotations for validation.

## 5 Retrieval Results

This section presents the results of testing 17 models discussed in Section 2 on the DORIS-MAE dataset. When available, we trained the model on our CS corpus and denoted the best version as trained in domain (ID), see full training details in Appendix D.1. To contextualize their performance on DORIS-MAE, we compare the results with these models' previously reported performances on various IR datasets, including MS MARCO [57], LoTTE [66], NQ [40], and Wiki-QA [86].

### 5.1 DORIS-MAE Benchmarking Results

In our benchmarking process for DORIS-MAE, we use complex queries as inputs to these models. We employ a variety of metrics common in the IR/NIR literature for the evaluation, including R@5, R@20, R-Precision (RP), $\text{NDCG}^{\text{exp}}_{10\%}$, MRR@10, and MAP. For fairness, we adopted an alternative approach for models like RocketQA-v2 [63] and ColBERT-v2 [66] that were not designed to handle long queries. For these cases, we allow the models to process the input as either a single text string or sentence-by-sentence, and report the maximum performance achieved. Uniquely among the models that we consider, the ASPIRE models (TSApire/OTAspire) are designed to handle multi-aspect queries. For brevity, we only report the higher number among these two options for models in Table 2. For more detailed results, refer to Appendix C.

To better interpret the results, we compare against a random ranking baseline. In general, the models show consistent behavior, with larger and more general-purpose models (like E5-Large-V2, RocketQA-v2, ada-002, Specter-v2) faring better than the smaller and more specialized ones (like SciBERT [5], ColBERT-v2, BM25, TF-IDF). Though the Aspire models were designed for multi-aspect queries, they do not have strong performance on the complex queries in DORIS-MAE.

When we compare the DORIS-MAE performance of these models with their reported results on traditional retrieval datasets MS MARCO (in Table 3) and NQ (in Table 4), we observe a significant reduction in their performances on DORIS-MAE. We choose metrics for comparison based on what is available in previously published work. The results highlight the challenges posed by DORIS-MAE and suggest gaps in the ability of existing methods to generalize well to complex query retrieval.

Finally, in Table 5, we make a comparison with the model performances on specialized retrieval datasets such as CSFCube [53], RELISH [10], and TRECCOVID [73]. The comparison reveals a

Table 2: Query level performance on full DORIS-MAE. Standard errors are estimated by bootstrapping. ID means a model is trained in domain.

| Method | R@5 | R@20 | RP | NDCG$_{10\%}^{\exp}$ | MRR@10 | MAP |
|---|---|---|---|---|---|---|
| Random | 4.41 | 18.48 | 16.29 | 7.31 | 3.59 | 19.63 |
| E5-L-v2[74] | **16.51** $\pm$ 2.05 | 43.77 $\pm$ 2.14 | **37.46** $\pm$ 2.44 | 25.90 $\pm$ 2.15 | 14.85 $\pm$ 2.73 | **40.49** $\pm$ 2.32 |
| RocketQA-v2[63] | 15.63 $\pm$ 1.88 | **45.41** $\pm$ 2.43 | 34.36 $\pm$ 2.32 | **30.30** $\pm$ 2.26 | **20.87** $\pm$ 3.12 | 40.18 $\pm$ 2.23 |
| ada-002[28] | 15.38 $\pm$ 1.95 | 42.84 $\pm$ 2.53 | 35.81 $\pm$ 2.67 | 27.46 $\pm$ 2.48 | 19.88 $\pm$ 3.21 | 40.37 $\pm$ 2.55 |
| SimCSE[25] | 14.90 $\pm$ 1.89 | 42.62 $\pm$ 2.40 | 35.27 $\pm$ 2.34 | 26.88 $\pm$ 2.36 | 21.19 $\pm$ 3.47 | 39.02 $\pm$ 2.35 |
| SPLADE-v2[25] | 14.78 $\pm$ 1.89 | 40.14 $\pm$ 2.33 | 31.65 $\pm$ 2.38 | 26.08 $\pm$ 2.00 | 17.82 $\pm$ 2.99 | 37.23 $\pm$ 2.26 |
| SPECTER-v2[18] | 14.50 $\pm$ 2.15 | 43.36 $\pm$ 2.50 | 33.41 $\pm$ 2.33 | 25.65 $\pm$ 2.23 | 17.19 $\pm$ 2.96 | 37.12 $\pm$ 2.10 |
| SPECTER$_{ID}$ | 13.32 $\pm$ 1.76 | 42.52 $\pm$ 2.37 | 31.55 $\pm$ 2.28 | 21.27 $\pm$ 2.03 | 14.48 $\pm$ 2.78 | 36.02 $\pm$ 2.19 |
| TSAspire[54] | 14.26 $\pm$ 1.80 | 41.25 $\pm$ 2.40 | 33.81 $\pm$ 2.47 | 26.63 $\pm$ 2.05 | 15.59 $\pm$ 2.59 | 37.00 $\pm$ 2.29 |
| SentBERT[62] | 14.09 $\pm$ 1.88 | 44.69 $\pm$ 2.47 | 33.79 $\pm$ 2.41 | 21.88 $\pm$ 2.07 | 13.23 $\pm$ 2.69 | 37.75 $\pm$ 2.28 |
| OTAspire[54] | 13.34 $\pm$ 1.56 | 42.25 $\pm$ 2.53 | 33.63 $\pm$ 2.38 | 25.52 $\pm$ 2.29 | 14.18 $\pm$ 2.66 | 36.70 $\pm$ 2.22 |
| ANCE$_{FirstP}$[85] | 13.21 $\pm$ 2.02 | 34.54 $\pm$ 2.20 | 30.51 $\pm$ 2.50 | 20.30 $\pm$ 2.02 | 13.87 $\pm$ 2.64 | 34.53 $\pm$ 2.35 |
| SPLADE-v2[23] | 11.80 $\pm$ 1.86 | 36.59 $\pm$ 2.12 | 29.90 $\pm$ 2.20 | 21.35 $\pm$ 2.12 | 14.30 $\pm$ 2.77 | 33.98 $\pm$ 2.23 |
| LLAMA[71] | 12.74 $\pm$ 1.82 | 34.51 $\pm$ 2.36 | 28.33 $\pm$ 2.14 | 16.65 $\pm$ 1.68 | 11.78 $\pm$ 2.45 | 31.29 $\pm$ 1.99 |
| SimLM[75] | 12.68 $\pm$ 1.77 | 35.67 $\pm$ 2.49 | 28.90 $\pm$ 2.42 | 18.91 $\pm$ 1.86 | 11.29 $\pm$ 2.44 | 33.06 $\pm$ 2.34 |
| BM25[72] | 8.47 $\pm$ 1.80 | 30.50 $\pm$ 2.38 | 21.94 $\pm$ 2.03 | 13.23 $\pm$ 1.97 | 9.19 $\pm$ 2.46 | 25.99 $\pm$ 1.68 |
| ColBERT-v2[66] | 8.45 $\pm$ 1.46 | 27.86 $\pm$ 2.29 | 22.33 $\pm$ 2.01 | 12.57 $\pm$ 1.71 | 6.69 $\pm$ 2.15 | 25.80 $\pm$ 1.83 |
| TF-IDF[67] | 10.71 $\pm$ 1.48 | 29.22 $\pm$ 2.25 | 24.79 $\pm$ 2.06 | 18.25 $\pm$ 2.01 | 12.41 $\pm$ 2.53 | 28.77 $\pm$ 1.81 |
| ERNIE[47] | 6.49 $\pm$ 0.94 | 22.58 $\pm$ 1.72 | 20.18 $\pm$ 1.82 | 9.66 $\pm$ 1.18 | 3.77 $\pm$ 1.06 | 22.71 $\pm$ 1.65 |
| SciBERT[5] | 5.13 $\pm$ 1.25 | 17.99 $\pm$ 1.69 | 17.13 $\pm$ 1.88 | 7.50 $\pm$ 1.34 | 3.41 $\pm$ 1.57 | 20.34 $\pm$ 1.64 |

consistent level of difficulty between DORIS-MAE and these completely human-annotated datasets, indicating that DORIS-MAE presents a similarly challenging retrieval task.

Table 3: Comparison with MS MARCO. Stats collected from [85, 66, 63, 75, 23].

| Ranking Method | MS MARCO MRR@10 | DORIS MRR@10 |
|---|---|---|
| ANCE | 33.0 | 13.87 |
| ColBERT-v2 | 39.7 | 6.69 |
| RocketQA-v2 | 41.9 | 20.87 |
| SimLM | 41.1 | 11.29 |
| SPLADE-v2 | 36.8 | 14.30 |

Table 4: Comparison with NQ. Stats collected from [66].

| Ranking Method | NQ R@20 | DORIS R@20 |
|---|---|---|
| ANCE$_{FirstP}$ | 81.9 | 34.54 |
| BM25 | 59.1 | 30.50 |
| RocketQA-v2 | 83.7 | 45.41 |
| SimLM | 85.2 | 35.67 |

Table 5: Comparison with CSFCube, TRECCOVID, and RELISH. Results are from [53, 54].

| Ranking Method | CSFCube RP | CSFCube MAP | CSFCube R@20 | TRECCOVID MAP | RELISH MAP | DORIS-MAE RP | DORIS-MAE MAP | DORIS-MAE R@20 |
|---|---|---|---|---|---|---|---|---|
| TSAspire | - | 40.26 | - | 26.24 | 61.29 | 33.81 | 37.00 | 41.25 |
| OTAspire | - | 40.79 | - | 30.92 | 62.57 | 33.63 | 36.70 | 42.25 |
| Specter-v2 | 18.32 | - | 52.12 | 28.24 | 60.62 | 33.41 | 37.12 | 43.36 |
| TF-IDF | 14.59 | - | 39.69 | - | - | 24.79 | 28.77 | 29.22 |
| BM25 | 13.50 | - | 42.73 | - | - | 21.94 | 25.99 | 30.50 |

## 5.2 Additional Experiments

Up until now, the hierarchical aspect-based structure that Anno-GPT utilizes has been hidden from all the evaluated models because of inability of existing methods to break down a complex query automatically. Even though these structures are not readily available for real-life retrieval methods, they may still hold value once this query decomposition process can be fully automated. To explore this potential, we conduct an experiment where instead of using the original query, each retrieval

method had access to a concatenated string of all aspects within a query, excluding sub-aspects. The results, as illustrated in Table 6, show that Sentence-BERT performs best on four metrics: R@5/20, RP, and MAP. These findings suggest that the use of aspect information could be potentially beneficial to guide retrieval methods.

The hierarchical structure of our dataset can be used to create additional, less complex tasks involving only parts of the query. For instance, by pulling out parts of the query corresponding to 2 Aspects, we are able to generate over 1000 test cases. For this task, we found a significant increase in the number of relevant abstracts. After re-evaluating all models for the sub-query DORIS-MAE, we observed model performance consistent with those seen in previous benchmarks, as indicated in Table 7. Comparing Table 7 with Table 2, we observed noticeably higher numbers for metrics such as RP, $NDCG_{10\%}^{exp}$, MRR and MAP, which are indicators of better model performances on this sub-query task. Since models remain unchanged, this suggest the sub-query task is less challenging than full-query task. This is intuitive since sub-queries are less complex and contain fewer aspects. Overall, the creation of these sub-query tasks underscores the adaptability of our dataset, which could accommodate a range of task complexities under the setting of DORIS-MAE.

Table 6: Ranking performance given model access to aspects. Full table is in Appendix C

| Method | R@5 | R@20 | RP | $NDCG_{10\%}^{exp}$ | MRR@10 | MAP |
|---|---|---|---|---|---|---|
| ada-002 | 14.09 | 42.23 | 33.56 | 26.54 | **20.20** | 37.62 |
| SentBERT | **17.73** | **45.34** | **35.67** | 25.00 | 15.52 | **39.87** |
| RocketQA-v2 | 13.83 | 43.81 | 32.59 | **27.45** | 16.08 | 37.90 |

Table 7: Ranking performance on sub-query (2 Aspects) task. Full table is in Appendix C

| Method | R@5 | R@20 | RP | $NDCG_{10\%}^{exp}$ | MRR@10 | MAP |
|---|---|---|---|---|---|---|
| ada-002 | **13.49** | **40.24** | **47.35** | **39.00** | **24.33** | **51.67** |
| SentBERT | 12.15 | 36.71 | 45.08 | 34.78 | 20.71 | 48.96 |
| RocketQA-v2 | 12.79 | 39.19 | 46.47 | 38.78 | 23.72 | 50.81 |

### 5.3 Supervised Learning on DORIS-MAE

To assess the utility of our dataset for training IR models, we conducted an experiment where we allocated 40 queries for training and the remaining 60 for testing. Using supervised contrastive learning (SCL), we optimized a margin-based triplet loss as presented in Equation 3. Each triplet, represented as $(A, P, N)$, consisted of a query and two abstracts. The higher-ranked abstract in the training data served as the positive instance (i.e., $P$), with the other functioning as the negative instance (i.e., $N$). Given each query's candidate pool size of $\geq 100$, we derived multiple triplets. This process yielded 3,000 triplets from the designated 40 training queries. Subsequently, we fine-tuned an E5-L-v2 model, a RoBERTa-based text embedding variant with 355 million parameters, over a single epoch.

$$L(A, P, N) = \max\left(\frac{<A, N>}{||A|| \cdot ||N||} - \frac{<A, P>}{||A|| \cdot ||P||} + m, 0\right) \quad A, P, N \in \mathbb{R}^n, m > 0 \quad (3)$$

When evaluating the model on the 60 test queries, we noted a marked improvement across all metrics with the SCL-trained model in contrast to the pre-trained baseline. This comparison can be found in Table 8. These positive outcomes, achieved using supervised contrastive learning on DORIS-MAE, underscore the utility of our train/test split for model fine-tuning.

Note that Table 2 reports models performance on the full DORIS-MAE dataset. We also report models performance on our proposed test set of 60 queries in Appendix C.4.

## 6 Conclusion and Future Work

This paper introduces a novel task, **S**cientific **DO**cument **R**etrieval using **M**ulti-level **A**spect-based qu**E**ries (DORIS-MAE), aimed at modeling the process of information retrieval in the context of

Table 8: Comparison of SCL vs pretrained. Standard errors are estimated by bootstrapping.

| Method | R@5 | R@20 | RP | NDCG$_{10\%}^{exp}$ | MRR@10 | MAP |
|---|---|---|---|---|---|---|
| SCL-trained E5-v2 | 19.57 $\pm$ 2.33 | 52.45 $\pm$ 3.17 | 44.47 $\pm$ 3.11 | 34.67 $\pm$ 3.17 | 23.16 $\pm$ 4.28 | 49.15 $\pm$ 3.14 |
| pretrained E5-v2 | 14.70 $\pm$ 1.72 | 42.38 $\pm$ 2.59 | 38.24 $\pm$ 2.94 | 26.31 $\pm$ 2.94 | 14.53 $\pm$ 3.69 | 40.62 $\pm$ 2.85 |

scientific research. We also present a dataset for DORIS-MAE generated using the Anno-GPT framework.

The results show room for improvement in the performance of current retrieval methods when dealing with DORIS-MAE. Future studies may explore modifications to model architectures and training procedures to better address complex, multifaceted queries.

An understanding of how noise in aspect annotation affects the overall task is an interesting point for future investigation, as it can shed light on how errors in the automated annotations may affect the final candidate pool ranking in DORIS-MAE.

The hierarchical structure of complex queries, as exemplified in DORIS-MAE, is an area that warrants further attention. Future work might include the development of more sophisticated automated query breakdown methods, potentially drawing from advances in question decomposition [36, 92], sequence-to-sequence modelling [42, 43] and semantic parsing [76, 88, 68].

**Limitations**: DORIS-MAE currently contains queries and abstracts from the computer science domain. Consequently, models trained on this dataset may not generalize well to other disciplines. An extended, multi-domain version of DORIS-MAE is a logical direction for future work. The task of determining aspect relevance is challenging due to the complexity of the abstracts. Improved annotation guidelines and training (for both humans and models) may address this challenge. Finally, while we harnessed LLMs to streamline the annotation, the generation of queries and their aspect decomposition remains manual. We found the development of a reliable, automated query generation pipeline difficult, but anticipate that advances in LLMs might bridge this gap in the near future.

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
