# Appendix: Table of Contents

# A Dataset

## A.1 Dataset License and Code License

The DORIS-MAE dataset is made available under the Creative Commons Attribution-NonCommercial (CC-BY-NC) license. This license allows users to share and adapt the dataset under the condition that they provide appropriate credit to the original creators and do not use it for commercial purposes. A copy of the full license can be found at https://github.com/Real-Doris-Mae/Doris-Mae-Dataset/blob/main/dataset_license.md.

The code used in this paper is released under the MIT License. The MIT License is a permissive open-source license that allows for the free use, modification, and distribution of the code, as long as the original license is included with any derivative work. A copy of the full license can be found at https://github.com/Real-Doris-Mae/Doris-Mae-Dataset/blob/main/code_license.md.

## A.2 Dataset Hosting, Accessibility and Maintenance

The DORIS-MAE dataset with its meta-data is released and can be accessed freely at (https://doi.org/10.5281/zenodo.8299749) under the CC-BY-NC license. We commit to regularly maintain the dataset and codebase by incorporating user feedback. We will potentially introduce more features as part of future work in the next version of DORIS-MAE. We confirm that the current version of DORIS-MAE would always remain accessible at the same link.

## A.3 Dataset Overview

The DORIS-MAE dataset is comprised of four main sub-datasets, each serving distinct purposes.

The **Query** dataset contains 100 human-crafted complex queries spanning across five categories: ML, NLP, CV, AI, and Composite. Each category has 20 associated queries. Queries are broken down into aspects (ranging from 3 to 9 per query) and sub-aspects (from 0 to 6 per aspect, with 0 signifying no further breakdown required). For each query, a corresponding candidate pool of relevant paper abstracts, ranging from 99 to 138, is provided.

The **Corpus** dataset is composed of 363,133 abstracts from computer science papers, published between 2011-2021, and sourced from arXiv. Each entry includes title, original abstract, URL, primary and secondary categories, as well as citation information retrieved from Semantic Scholar. A masked version of each abstract is also provided, facilitating the automated creation of queries.

The **Annotation** dataset includes generated annotations for all 165,144 question pairs, each comprising an aspect/sub-aspect and a corresponding paper abstract from the query's candidate pool. It includes the original text generated by ChatGPT explaining its decision-making process, along with a three-level relevance score (e.g., 0,1,2) representing ChatGPT's final decision.

Finally, the **Test Set** dataset contains human annotations for a random selection of 250 question pairs used in hypothesis testing. It includes each of the three human annotators' final decisions, recorded as a three-level relevance score (e.g., 0,1,2). Note, the development set contains 90 annotated question pairs. Since the annotations in the development set come from only 2 annotators and since the development set is strictly not a part of the hypothesis testing stage, we do not include the development set in the officially released DORIS-MAE dataset. Instead, we provide the development set as part of the codebase on https://github.com/Real-Doris-Mae/Doris-Mae-Dataset/.

It is important to note that each of these datasets serves a specific function in our overall process, contributing to the optimization, validation, and benchmarking of our proposed approach.

## A.4 DORIS-MAE Structure

- `aspect2aspect_id`: A dictionary that maps aspects/sub-aspects to their aspect_id.
- `aspect_id2aspect`: The inverse mapping of `aspect2aspect_id`.
- **Query**: contains 50 complex queries.
  - Query 1:
    - `query_text`: The text of the query.

- query_type: One of the [ML, NLP, CV, AI, Composite].
- idea_from: The id of the referenced abstract when creating the query
- candidate_pool: The list of abstract_ids in the candidate pool of Query 1.
- sent2aspect_id: A dictionary that maps individual sentence to its aspect_id(s).
- aspect_id2sent: The inverse mapping of sent2aspect_id.
- aspects: contains all aspects ids excluding sub-aspects for Query 1.
  • aspect1_id: [sub-aspect1_id, sub-aspect2_id,...]
  • aspect2_id:
  • ...
• Query 2:
• ...
- **Corpus**: contains 363,133 Computer Science(CS) paper abstracts.
  • Abstract 1:
    - original_abstract: The text of the abstract.
    - masked_abstract: The text of the abstract with specialized topic words masked.
    - title: Title of the paper.
    - url: Link to the paper.
    - primary_category: Primary category of the paper, assigned by arXiv.
    - categories: Full list of the categories of the paper, assigned by arXiv.
    - ss_id: The paper's semantic scholar id.
    - incoming_citations: List of abstract_ids citing this paper.
    - outgoing_citations: List of abstract_ids cited by this paper.
    - abstract_id: the index 1 is the abstract_id for this paper.
  • Abstract 2:
  • ...
- **Annotation**: contains 83,591 annotated question pairs.
  • Annotation 1: The first question pair.
    - aspect_id: The id of the aspect/sub-aspect in the question pair.
    - abstract_id: The id of the abstract in the question pair.
    - gpt_response: The reasoning and decision by ChatGPT for this question pair.
    - score: The relevance score of the question pair (e.g. 0,1,2).
  • Annotation 2:
  • ...
- **Test Set**: contains 250 humanly annotated question pairs.
  • Test Set Question Pair 1:
    - aspect_id: The id of the aspect/sub-aspect in the question pair.
    - abstract_id: The id of the abstract in the question pair.
    - human_annotation: contains the relevance score assigned by all 3 annotators.
      • annotator_1: Score of annotator_1.
      • annotator_2: Score of annotator_2.
      • annotator_3: Score of annotator_3.
  • Test Set Question Pair 2:
  • ...

## A.5 Statistics of Dataset

To better interpret the results of our evaluations and guide future improvements, we provide several analyses of the distribution and relevance of the abstracts in our DORIS-MAE dataset.

We used Equations 1 and 2 to determine the relevance of an abstract to a specific query. We then examined the distribution of the number of relevant abstracts for different queries and considered how this might impact various evaluation metrics.

Table 9 presents the distribution of relevant abstracts categorized by query type (ML, NLP, CV, AI, and Composite) in our dataset. For each category, we determined the count of relevant abstracts for

Table 9: Number of relevant abstracts for each query type. Comp means composite queries. Sub-query stands for two-aspect sub-queries.

| | | Min | Max | Mean | Std |
|---|---|---|---|---|---|
| | ML | 2 | 84 | 24.35 | 19.47 |
| | NLP | 0 | 41 | 17.65 | 11.98 |
| Query | CV | 0 | 52 | 18.55 | 13.29 |
| | AI | 1 | 51 | 11.60 | 12.35 |
| | Comp | 0 | 61 | 9.95 | 14.63 |
| | All | 0 | 84 | 16.42 | 15.49 |
| | ML | 1 | 105 | 36.40 | 25.34 |
| | NLP | 0 | 83 | 29.86 | 18.56 |
| Sub-query | CV | 1 | 99 | 28.52 | 16.57 |
| | AI | 0 | 79 | 20.96 | 16.16 |
| | Comp | 0 | 81 | 23.09 | 21.91 |
| | All | 0 | 105 | 28.14 | 21.07 |

Table 10: Normalized score of abstracts for each query type. Compo means composite queries. Sub-query stands for two-aspect sub-queries.

| | | Min | Max | Mean | Std |
|---|---|---|---|---|---|
| | ML | 0 | 1.778 | 0.665 | 0.377 |
| | NLP | 0 | 1.786 | 0.609 | 0.361 |
| Query | CV | 0 | 1.615 | 0.623 | 0.339 |
| | AI | 0 | 1.667 | 0.510 | 0.337 |
| | Comp | 0 | 1.909 | 0.483 | 0.332 |
| | All | 0 | 1.909 | 0.578 | 0.356 |
| | ML | 0 | 2.000 | 0.679 | 0.478 |
| | NLP | 0 | 2.000 | 0.638 | 0.453 |
| Sub-query | CV | 0 | 2.000 | 0.642 | 0.434 |
| | AI | 0 | 2.000 | 0.533 | 0.419 |
| | Comp | 0 | 2.000 | 0.544 | 0.425 |
| | All | 0 | 2.000 | 0.610 | 0.448 |

its respective queries and provided summary statistics of these counts. We find that composite queries have a lower average number of relevant abstracts, implying that such queries might encompass aspects or sub-aspects not extensively covered by papers in their associated pools. This observation, along with the varied number of relevant abstracts per query (ranging from 0 to 84), suggests that the DORIS-MAE dataset encompasses queries of diverse difficulty levels. It's noteworthy that four queries lack any relevant abstracts and were consequently omitted from our evaluation.

The lower portion of Table 9 presents the distribution of relevant abstracts for 1003 sub-queries with two aspects. As the table shows, the number of relevant abstracts increases across all five types, suggesting that the sub-query DORIS-MAE task is less challenging than the full-query DORIS-MAE task.

Next, we examined the distribution of normalized scores (defined as $\frac{S(p_j|Q)}{|Q|}$) for each abstract given a query. We compiled the normalized scores for each abstract in the candidate pools for each query type and report the aggregate statistics in Table 10. Composite queries have the lowest average relevance scores.

Figure 2(a) shows the normalized scores within a candidate pool (ranked in descending order) for a given query. Figure 2(b) aggregates the data from Figure 2(a). As seen in Figure 2(a), there are only a few plateaus (i.e., abstracts with tied scores), which means the normalized scores provide a fine-grained ranking for the candidate pool. This granularity is valuable when evaluating reranking models.

# B  Annotation

## B.1  Annotation Guidelines

In Section 4, we introduced the grading scale (0-2) for evaluating the relationship between an abstract and an aspect/sub-aspect in a question pair. To facilitate this annotation process, we expand the definition of Level 1 into two sub-categories, creating a four-level grading scale during the annotation phase. Afterward, these levels are mapped back onto the original three-level scale.

The four-level grading scale and their corresponding mappings are as follows:

1. **DISAGREE**: This corresponds to Level 0 in the main paper. The abstract is unrelated or does not provide any help to the key components of the aspect or sub-aspect.

2. **DISPUTE**: This is a sub-category of Level 1. The abstract answers some key components and is adaptable to fulfill the aspect/sub-aspect. However, the abstract does not directly address the aspect/sub-aspect.

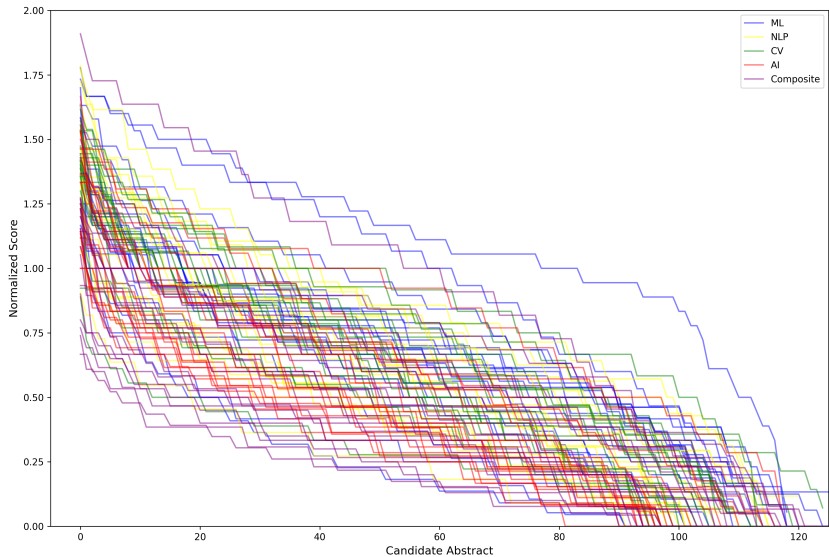

(a) Normalized scores of abstracts in candidate pool per query.

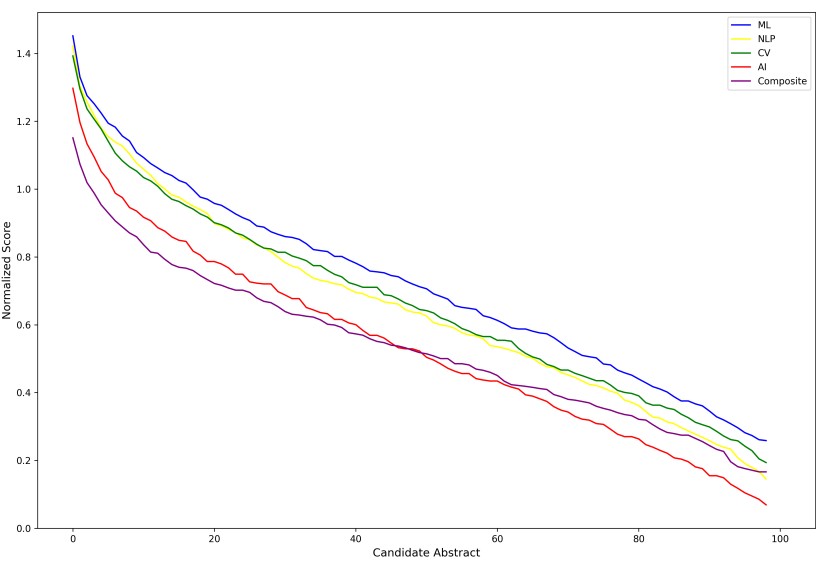

(b) Aggregated normalized scores of abstracts per query type.

Figure 2: Each line in (a) represents a query. Each line in (b) represents a query type. The color of the line represents the type of the query. The candidate abstracts are in descending order w.r.t. their normalized scores.

3. **AGREE**: This is another sub-category of Level 1. The abstract directly addresses only a portion of the aspect/sub-aspect, but not in its entirety.

4. **CONCUR**: This corresponds to Level 2. The abstract directly answers all the key components of the aspect or sub-aspect.

These labels "DISAGREE", "DISPUTE", "AGREE", and "CONCUR" serve to add semantic meaning to the grading levels, aiding annotators during the annotation process.

For consistency, "DISPUTE" and "AGREE" are treated as equal in weight, representing different scenarios under Level 1, but neither is considered stronger or weaker than the other.

During the annotation process, annotators must follow a set of guidelines designed to maintain consistency and minimize bias. These guidelines cover aspects like time management, communication among annotators (which is explicitly forbidden), use of additional resources, the process of question analysis, and the identification of key components. Details of these guidelines can be found in the following section.

1. **Resource usage**: Given the technical nature of the abstracts, annotators may occasionally need to look up unfamiliar terms. They can use traditional search engines for these instances. However, they are strictly prohibited from using any large language model (LLM) interfaces or search engines powered by LLMs.

2. **Annotation process**: Annotators are required to read the entire abstract and question at least once before making a decision. They need to refresh their understanding of the four grading levels in their minds before finalizing their decision. On average, this process takes about 4 minutes, but no strict time limit is enforced. During the annotation process, annotators are strictly and explicitly prohibited from communicating in any way.

3. **Key component identification**: The grading scale revolves around the concept of "key components". It's up to the annotators, leveraging their research experience, to identify what constitutes as a key component in an aspect/sub-aspect. If there are multiple key components (e.g., 2 or 3), they are expected to monitor all of these components and verify whether the abstract mentions them.

4. **Inference rules**: Annotators can leverage their expert knowledge to draw reasonable inferences from the abstract's content with respect to the key components. However, they should avoid unnecessary complications in the correlation between the aspect and the abstract. Each inference made must have a clear, straightforward justification. The aim here is to ensure accuracy without over-complicating the annotation process.

## B.2 Examples for Annotations

In this section, we provide examples of aspect/abstract pairs with different relevance scores.

### B.2.1 Examples with relevance score 0

*Aspect/Sub-aspect*: The paper mentions the possibility that the labeling process introduces unwanted randomness or noise.

*Abstract*: Recent approaches in literature have exploited the multi-modal information in documents (text, layout, image) to serve specific downstream document tasks. However, they are limited by their - (i) inability to learn cross-modal representations across text, layout and image dimensions for documents and (ii) inability to process multi-page documents. Pre-training techniques have been shown in Natural Language Processing (NLP) domain to learn generic textual representations from large unlabelled datasets, applicable to various downstream NLP tasks. In this paper, we propose a multi-task learning-based framework that utilizes a combination of self-supervised and supervised pre-training tasks to learn a generic document representation applicable to various downstream document tasks. Specifically, we introduce Document Topic Modelling and Document Shuffle Prediction as novel pre-training tasks to learn rich image representations along with the text and layout representations for documents. We utilize the Longformer network architecture as the backbone to encode the multi-modal information from multi-page documents in an end-to-end fashion. We showcase the applicability of our pre-training framework on a variety of different real-world document tasks such as document classification, document information extraction, and document retrieval. We evaluate our framework on different standard document datasets and

conduct exhaustive experiments to compare performance against various ablations of our framework and state-of-the-art baselines.

*Human reasoning*: The above abstract is not related to the aspect.

. . . . . . . . . . . . . . . . . . . . . . . . . . . . . . . . . . . . . . . . . . . . . . . . . . . . . . . . . . . . . . . . . . . . . . . . . . . . . . . . . . . . . . . . . .

*Aspect/Sub-aspect*: The document should use semi-definite programming to estimate error bound.

*Abstract*: We develop methodology for estimation and inference using machine learning to enrich economic models. Our framework takes a standard economic model and recasts the parameters as fully flexible nonparametric functions, to capture the rich heterogeneity based on potentially high dimensional or complex observable characteristics. These "parameter functions" retain the interpretability, economic meaning, and discipline of classical parameters. Deep learning is particularly well-suited to structured modeling of heterogeneity in economics. We show how to design the network architecture to match the structure of the economic model, delivering novel methodology that moves deep learning beyond prediction. We prove convergence rates for the estimated parameter functions. These functions are the key inputs into the finite-dimensional parameter of inferential interest. We obtain inference based on a novel influence function calculation that covers any second-stage parameter and any machine-learning-enriched model that uses a smooth per-observation loss function. No additional derivations are required. The score can be taken directly to data, using automatic differentiation if needed. The researcher need only define the original model and define the parameter of interest. A key insight is that we need not write down the influence function in order to evaluate it on the data. Our framework gives new results for a host of contexts, covering such diverse examples as price elasticities, willingness-to-pay, and surplus measures in binary or multinomial choice models, effects of continuous treatment variables, fractional outcome models, count data, heterogeneous production functions, and more. We apply our methodology to a large scale advertising experiment for short-term loans. We show how economically meaningful estimates and inferences can be made that would be unavailable without our results.

*Human reasoning*: The above abstract is not related to the aspect.

### B.2.2 Examples with relevance score 1

*Aspect/Sub-aspect*: The paper should introduce a system that can represent the possible options and consequences in decision making using actions and states.

*Abstract*: Decision-making is often dependent on uncertain data, e.g. data associated with confidence scores or probabilities. We present a comparison of different information presentations for uncertain data and, for the first time, measure their effects on human decision-making. We show that the use of Natural Language Generation (NLG) improves decision-making under uncertainty, compared to state-of-the-art graphical-based representation methods. In a task-based study with 442 adults, we found that presentations using NLG lead to 24% better decision-making on average than the graphical presentations, and to 44% better decision-making when NLG is combined with graphics. We also show that women achieve significantly better results when presented with NLG output (an 87% increase on average compared to graphical presentations).

*Human reasoning*: The above abstract explicitly discuss decision-making. However, the abstract does not mention anything about actions and states. Thus, some key components are missing. More specifically, it belongs to the class "Agree" under Level 1.

. . . . . . . . . . . . . . . . . . . . . . . . . . . . . . . . . . . . . . . . . . . . . . . . . . . . . . . . . . . . . . . . . . . . . . . . . . . . . . . . . . . . . . . . . .

*Aspect/Sub-aspect*: The document should introduce a system that can retrieve fine-grained video of actions from longer videos and match these actions to the

text description or instruction.

*Abstract*: In this paper, we propose a new approach for retrieval of video segments using natural language queries. Unlike most previous approaches such as concept-based methods or rule-based structured models, the proposed method uses image captioning model to construct sentential queries for visual information. In detail, our approach exploits multiple captions generated by visual features in each image with 'Densecap'. Then, the similarities between captions of adjacent images are calculated, which is used to track semantically similar captions over multiple frames. Besides introducing this novel idea of 'tracking by captioning', the proposed method is one of the first approaches that uses a language generation model learned by neural networks to construct semantic query describing the relations and properties of visual information. To evaluate the effectiveness of our approach, we have created a new evaluation dataset, which contains about 348 segments of scenes in 20 movie-trailers. Through quantitative and qualitative evaluation, we show that our method is effective for retrieval of video segments using natural language queries.

*Human reasoning*: The above abstract mentions retrieval of video segments using natural language queries, which is related to the aspect. However, another key component of the abstract is "action". Although it is very natural to infer that the system in the document can retrieve video of actions, it is not explicitly mentioned in the abstract. More specifically, it belongs to the class "Dispute" under Level 1.

### B.2.3   Examples with relevance score 2

*Aspect/Sub-aspect*: The paper should mention chemistry or chemistry related knowledge.

*Abstract*: Modern astronomical surveys are observing spectral data for millions of stars. These spectra contain chemical information that can be used to trace the Galaxy's formation and chemical enrichment history. However, extracting the information from spectra, and making precise and accurate chemical abundance measurements are challenging. Here, we present a data-driven method for isolating the chemical factors of variation in stellar spectra from those of other parameters (i.e. \teff, \logg, eh). This enables us to build a spectral projection for each star with these parameters removed. We do this with no ab initio knowledge of elemental abundances themselves, and hence bypass the uncertainties and systematics associated with modeling that rely on synthetic stellar spectra. To remove known non-chemical factors of variation, we develop and implement a neural network architecture that learns a disentangled spectral representation. We simulate our recovery of chemically identical stars using the disentangled spectra in a synthetic APOGEE-like dataset. We show that this recovery declines as a function of the signal to noise ratio, but that our neural network architecture outperforms simpler modeling choices. Our work demonstrates the feasibility of data-driven abundance-free chemical tagging.

*Human reasoning*: The above abstract explicitly mention "recovery of chemically identical stars" which is related to chemistry.

..........................................................................................

*Aspect/Sub-aspect*: The paper should provide robust safeguards to sensitive patient data.

*Abstract*: In survival analysis, regression models are used to understand the effects of explanatory variables (e.g., age, sex, weight, etc.) to the survival probability. However, for sensitive survival data such as medical data, there are serious concerns about the privacy of individuals in the data set when medical data is used to fit the regression models. The closest work addressing such privacy concerns is the work on Cox regression which linearly projects the original data to a lower dimensional space. However, the weakness of this approach is that there is

no formal privacy guarantee for such projection. In this work, we aim to propose solutions for the regression problem in survival analysis with the protection of differential privacy which is a golden standard of privacy protection in data privacy research. To this end, we extend the Output Perturbation and Objective Perturbation approaches which are originally proposed to protect differential privacy for the Empirical Risk Minimization (ERM) problems. In addition, we also propose a novel sampling approach based on the Markov Chain Monte Carlo (MCMC) method to practically guarantee differential privacy with better accuracy. We show that our proposed approaches achieve good accuracy as compared to the non-private results while guaranteeing differential privacy for individuals in the private data set.

*Human reasoning*: The above abstract explicitly discuss differential privacy" and sensitive private medical data, which satisfies all components in the abstract.

## B.3 Prompt Engineering

The prompt strategy was optimized on the development set, and was designed with an emphasis on clarity and conciseness. It aimed to elicit detailed reasoning and a definitive final decision from ChatGPT.

Key to the final prompt design was the selection of labels for the grading options: "Disagree," "Dispute," "Agree," and "Concur." These labels were chosen to reflect the different grading scale levels defined in Section B.1.

These labels, along with their detailed definitions, are included in the prompt. They serve to guide ChatGPT in its decision-making process by providing consistent terminology for each grading level. The final prompt used in the hypothesis testing stage is displayed below.

```
You are a reference librarian.  A user wants to find a
scientific paper that satisfies a particular requirement.
Based on the abstract of the paper, you will determine whether
the requirement is met.

Here is the user requirement:  {aspect}
Here is the abstract:  {abstract}

Explain whether the abstract satisfies the user requirement.
Think step by step, showing all of your reasoning.  Use the
following *rules* to determine whether the requirement is
satisfied:

Disagree means the abstract does not address the user
requirement;
Dispute means the abstract does not directly address the user
requirement, but is adaptable to fulfill the user's need;
Concur means the abstract directly and completely addresses
the user requirement;
Agree means the abstract directly addresses only a portion of
the user requirement, but not in its entirety.

Think carefully about the aforementioned *rules* to decide
whether you [DISAGREE, DISPUTE, AGREE, CONCUR] based on the
definition of these 4 options.  After you have provided your
reasoning, state your final selection in *all capital* letters
at the *end* of your response.
```

This prompt produced the best results on the development set, with the distribution of its responses closely matching that of human annotators. Based on these results, we proceeded to use this prompt in the testing process.

### B.4 Error Analysis for ChatGPT's Annotation

Though the final prompt yielded results comparable to human annotators, we aimed to identify potential inconsistencies in ChatGPT's responses that slightly deviate from humans' answers. We have organized them into several categories:

#### B.4.1 Definition Inconsistency

Test Case Prompt:

> You are a reference librarian. A user wants to find a scientific paper that satisfies a particular requirement. Based on the abstract of the paper, you will determine whether the requirement is met.
>
> Here is the user requirement: The model in the document should have the ability to discover the geometry and global structure of the face
>
> Here is the abstract: Facial image super-resolution (SR) is an important preprocessing for facial image analysis, face recognition, and image-based 3D face reconstruction. Recent convolutional neural network (CNN) based method has shown excellent performance by learning mapping relation using pairs of low-resolution (LR) and high-resolution (HR) facial images. However, since the HR facial image reconstruction using CNN is conventionally aimed to increase the PSNR and SSIM metrics, the reconstructed HR image might not be realistic even with high scores. An adversarial framework is proposed in this study to reconstruct the HR facial image by simultaneously generating an HR image with and without blur. First, the spatial resolution of the LR facial image is increased by eight times using a five-layer CNN. Then, the encoder extracts the features of the up-scaled image. These features are finally sent to two branches (decoders) to generate an HR facial image with and without blur. In addition, local and global discriminators are combined to focus on the reconstruction of HR facial structures. Experiment results show that the proposed algorithm generates a realistic HR facial image. Furthermore, the proposed method can generate a variety of different facial images.
>
> Explain whether the abstract satisfies the user requirement. Think step by step, showing all of your reasoning. Use the following *rules* to determine whether the requirement is satisfied: disagree means the abstract does not address the user requirement; dispute means the abstract does not directly address the user requirement, but is adaptable to fulfill the user's need; concur means the abstract directly and completely addresses the user requirement; agree means the abstract directly addresses only a portion of the user requirement, but not in its entirety. Think carefully about the aforementioned *rules* to decide whether you [DISAGREE, DISPUTE, AGREE, CONCUR] based on the definition of these 4 options. After you have provided your reasoning, state your final selection in *all capital* letters at the *end* of your response.

ChatGPT's Reasoning:

> The user requirement is that the model in the document should have the ability to discover the geometry and global structure of the face. The abstract discusses a proposed algorithm for facial image super-resolution using an adversarial framework that generates a realistic HR facial image. The algorithm uses local and global discriminators to focus on the reconstruction of HR facial structures. Based on this information, I CONCUR that the abstract satisfies the user requirement. The proposed algorithm is designed to reconstruct the HR facial image by simultaneously generating an HR image with and without blur. The local and global discriminators are combined to focus on the reconstruction of HR facial structures, which implies that the model has the ability to discover the geometry and global structure of the face. Therefore, the abstract directly and completely addresses the user requirement.

**Error Analysis:** Despite the abstract's implied ability to discern facial structures, it doesn't state this explicitly. However, even after recognizing the ability as "implied," ChatGPT marked the abstract

as directly satisfying the requirement. This indicates occasional misalignment between ChatGPT's reasoning and the labeling criteria.

### B.4.2 Inference Inconsistency

Test Case Prompt:

> You are a reference librarian. A user wants to find a scientific paper that satisfies a particular requirement. Based on the abstract of the paper, you will determine whether the requirement is met.
>
> Here is the user requirement: The document should introduce a model that can take human faces with stripe patterns and accurately remove the stripe pattern.
>
> Here is the abstract: In this paper, we propose a robust 3D face recognition system which can handle pose as well as occlusions in real world. The system at first takes as input, a 3D range image, simultaneously registers it using ICP(Iterative Closest Point) algorithm. ICP used in this work, registers facial surfaces to a common model by minimizing distances between a probe model and a gallery model. However the performance of ICP relies heavily on the initial conditions. Hence, it is necessary to provide an initial registration, which will be improved iteratively and finally converge to the best alignment possible. Once the faces are registered, the occlusions are automatically extracted by thresholding the depth map values of the 3D image. After the occluded regions are detected, restoration is done by Principal Component Analysis (PCA). The restored images, after the removal of occlusions, are then fed to the recognition system for classification purpose. Features are extracted from the reconstructed non-occluded face images in the form of face normals. The experimental results which were obtained on the occluded facial images from the Bosphorus 3D face database, illustrate that our occlusion compensation scheme has attained a recognition accuracy of 91.30
>
> Explain whether the abstract satisfies the user requirement. Think step by step, showing all of your reasoning. Use the following *rules* to determine whether the requirement is satisfied: disagree means the abstract does not address the user requirement; dispute means the abstract does not directly address the user requirement, but is adaptable to fulfill the user's need; concur means the abstract directly and completely addresses the user requirement; agree means the abstract directly addresses only a portion of the user requirement, but not in its entirety. Think carefully about the aforementioned *rules* to decide whether you [DISAGREE, DISPUTE, AGREE, CONCUR] based on the definition of these 4 options. After you have provided your reasoning, state your final selection in *all capital* letters at the *end* of your response.

ChatGPT's Reasoning:

> The abstract DISAGREES with the user requirement. The paper proposes a 3D face recognition system that can handle pose and occlusions in real-world scenarios. The system takes a 3D range image as input, registers it using the ICP algorithm, and extracts occlusions by thresholding the depth map values of the 3D image. However, the paper does not introduce a model that can remove stripe patterns from human faces. Therefore, the abstract does not address the user requirement.

**Error Analysis:** The human annotators labeled this question as "Dispute" due to the mention of occlusion detection, removal, and restoration in the abstract, inferring that "stripe patterns" could fall under the broader category of "occlusions". However, ChatGPT did not make the same inference, indicating a different style of reasoning.

### B.4.3 Key Component Inconsistency

Test Case Prompt:

You are a reference librarian. A user wants to find a scientific paper that satisfies a particular requirement. Based on the abstract of the paper, you will determine whether the requirement is met.

Here is the user requirement: The navigation agent model in the document should have the ability to be trained on data with varying levels of augmentation

Here is the abstract: Localization is the problem of estimating the location of an autonomous agent from an observation and a map of the environment. Traditional methods of localization, which filter the belief based on the observations, are sub-optimal in the number of steps required, as they do not decide the actions taken by the agent. We propose "Active Neural Localizer", a fully differentiable neural network that learns to localize accurately and efficiently. The proposed model incorporates ideas of traditional filtering-based localization methods, by using a structured belief of the state with multiplicative interactions to propagate belief, and combines it with a policy model to localize accurately while minimizing the number of steps required for localization. Active Neural Localizer is trained end-to-end with reinforcement learning. We use a variety of simulation environments for our experiments which include random 2D mazes, random mazes in the Doom game engine and a photo-realistic environment in the Unreal game engine. The results on the 2D environments show the effectiveness of the learned policy in an idealistic setting while results on the 3D environments demonstrate the model's capability of learning the policy and perceptual model jointly from raw-pixel based RGB observations. We also show that a model trained on random textures in the Doom environment generalizes well to a photo-realistic office space environment in the Unreal engine.

Explain whether the abstract satisfies the user requirement. Think step by step, showing all of your reasoning. Use the following *rules* to determine whether the requirement is satisfied: disagree means the abstract does not address the user requirement; dispute means the abstract does not directly address the user requirement, but is adaptable to fulfill the user's need; concur means the abstract directly and completely addresses the user requirement; agree means the abstract directly addresses only a portion of the user requirement, but not in its entirety. Think carefully about the aforementioned *rules* to decide whether you [DISAGREE, DISPUTE, AGREE, CONCUR] based on the definition of these 4 options. After you have provided your reasoning, state your final selection in *all capital* letters at the *end* of your response.

ChatGPT's Reasoning:

The abstract does not directly address the user requirement. The abstract discusses a neural network model for localization of an autonomous agent in various environments. The model is trained with reinforcement learning and is able to learn the policy and perceptual model jointly from raw-pixel based RGB observations. However, there is no mention of the navigation agent model being trained on data with varying levels of augmentation. Therefore, I DISAGREE that the abstract satisfies the user requirement.

**Error Analysis:**   Human annotator 1 assigned the label "Dispute" and human annotator 2 assigned the label "Disagree", while ChatGPT assigned the label "Disagree". The difference between ChatGPT and human annotator 1 in this case comes from their different definition of key components in the aspect. Human annotator 1 believes the navigation agent model itself is a key component, thus by mentioning the localization problem of an autonomous agent, the abstract already indirectly implies the navigation agent model. However, both ChatGPT and annotator 2 believes navigation agent that can be trained on data with varying levels of augmentation is the only key component in the aspect. Therefore, since data with varying level of augmentation is not mentioned in the abstract, both of them assigned "Disagree" indicating the abstract is unrelated to the only key component in the aspect. This form of inconsistency is the most common form of inconsistencies we found in ChatGPT's reasoning versus some human's reasoning. However, since humans also disagree on what constitutes as key components occasionally, this form of key component inconsistency cannot be completely avoided even when ChatGPT exhibits comparable performance as humans.

### B.5 Annotation Scalability

The Anno-GPT framework, as detailed in Section 4, employs ChatGPT for automated annotations, contingent on its performance being comparable to human experts during the hypothesis testing stage. This system scales linearly with the number of complex queries, aspects/sub-aspects, and the size of candidate pools per query, enabling larger datasets for training and testing purposes. For instance, with ten API keys, a multi-threaded program could create a training-sized dataset of 10,000 complex queries, each with a candidate pool of 100 abstracts, in approximately 240 hours.

Creating a DORIS-MAE dataset for another field, such as quantum computing or biomedical engineering, involves several preparatory stages. Initial complex queries (e.g., 50 queries) need to be either expertly crafted or gathered from online technical discussions in the field. Next, the complex queries must be broken down into a list of aspects and sub-aspects. This process could be carried out by an expert in the field or a proficient annotator.

Finally, it is critical to fine-tune the final prompt and verify ChatGPT's competence in the new field of research. As per our annotation guidelines, this stage involves three expert annotators, each dedicating an average of 20 hours. This step constitutes the largest fixed cost of applying Anno-GPT to a new field, but it's a necessary investment for ensuring dataset quality. However, once a language model such as ChatGPT is confirmed to perform at a comparable level to human experts, the Anno-GPT framework can then be used to create larger datasets in the domain.

## C  Retrieval Results

This section presents the full results of benchmarking sixteen models (ada-002 [28], E5-Large-v2 [74], LLAMA [71], Text-Supervised (TS)-ASPIRE [54], Optimal-Transport (OT)-ASPIRE [54], SPECTER [18], Sentence-BERT [62], RocketQA-v2 [63], ANCE FirstP [85], SimLM [75], SPLADE-v2 [23], ColBERT-v2 [66], SciBERT [5], ERNIE-Search [47], BM25 [72], TF-IDF [67]) discussed in Section 2 on the DORIS-MAE dataset. For SciBERT and SPECTER, we trained them on our 360k CS corpus with citation signals (i.e. in domain) and denoted their best trained version as SciBERT$_{ID}$ and SPECTER$_{ID}$.

### C.1  Metrics

In our experiments, we used several metrics which are commonly used in the IR/NIR literature for evaluating ranking and retrieval models. These include Recall@5 (R@5), Recall@20 (R@20), R-Precision (RP), NDCG$_{10\%}$, NDCG$_{10\%}^{\exp}$ [12], Mean Reciprocal Rank@10 (MRR@10), and Mean Average Precision (MAP). An abstract was categorized as relevant when its score from Equation 1 $\frac{S(p_j|Q)}{|Q|} \geq 1$. Note that NDCG$_{10\%}$ is excluded from the main paper due to an atypically high random baseline value of 48.71%.

### C.2  Query Level Benchmark Results

### C.2.1  Whole Query Embedding

In Table 11, we showcase the results obtained when models (ada-002, E5-Large-v2, LLAMA, SPECTER, SentBERT, RocketQA-v2, ANCE, SPLADE-v2, ColBERT-v2, SciBERT, ERNIE, BM25, TF-IDF) embed the entire query as a single vector and embed the entire abstract as a single vector. The relevance between the query and the abstract is determined by calculating the cosine similarity or L2 distance (with SPECTER using L2 distance as recommended by Cohan et al. [18]) between the query vector and the abstract vector.

In Table 11, we see the general-purposed text embedding models (ada-002, E5-Large-v2, LLAMA, SPECTER) (that are not restricted by the size of context window) typically outperform specialized NIR models (RocketQA-v2, SimLM, SPLADE-v2, ColBERT-v2). In-domain fine-tuning improves performance for SciBERT but shows mixed results for SPECTER.

Table 11: Models' query-level ranking performance (embedding query and abstract as single vectors)

| Method | R@5 | R@20 | RP | $\text{NDCG}_{10\%}$ | $\text{NDCG}^{\text{exp}}_{10\%}$ | MRR@10 | MAP |
|---|---|---|---|---|---|---|---|
| random | 4.41 | 18.48 | 16.29 | 48.71 | 7.31 | 3.59 | 19.63 |
| ada-002 | $15.38 \pm 1.95$ | $42.84 \pm 2.53$ | $35.81 \pm 2.67$ | $\mathbf{72.48} \pm 1.10$ | $\mathbf{27.46} \pm 2.48$ | $\mathbf{19.88} \pm 3.21$ | $40.37 \pm 2.55$ |
| E5-L-v2 | $\mathbf{16.51} \pm 2.05$ | $\mathbf{43.77} \pm 2.14$ | $\mathbf{37.46} \pm 2.44$ | $70.11 \pm 0.97$ | $25.90 \pm 2.15$ | $14.85 \pm 2.73$ | $\mathbf{40.49} \pm 2.32$ |
| LLAMA | $12.74 \pm 1.82$ | $34.51 \pm 2.36$ | $28.33 \pm 2.14$ | $64.75 \pm 1.04$ | $16.65 \pm 1.68$ | $11.78 \pm 2.45$ | $31.29 \pm 1.99$ |
| SPLADE-v2[25] | $14.78 \pm 1.89$ | $40.14 \pm 2.33$ | $31.65 \pm 2.38$ | $71.29 \pm 0.89$ | $26.08 \pm 2.00$ | $17.82 \pm 2.99$ | $37.23 \pm 2.26$ |
| SPECTER-v2 | $14.50 \pm 2.15$ | $43.36 \pm 2.50$ | $33.41 \pm 2.33$ | $70.19 \pm 0.98$ | $25.65 \pm 2.23$ | $17.19 \pm 2.96$ | $37.12 \pm 2.10$ |
| SPECTER | $13.34 \pm 1.92$ | $40.25 \pm 2.60$ | $32.66 \pm 2.55$ | $68.06 \pm 1.15$ | $24.43 \pm 2.35$ | $18.23 \pm 2.99$ | $35.75 \pm 2.46$ |
| $\text{SPECTER}_{\text{ID}}$ | $13.32 \pm 1.76$ | $42.52 \pm 2.37$ | $31.55 \pm 2.28$ | $69.34 \pm 1.09$ | $21.27 \pm 2.03$ | $14.48 \pm 2.78$ | $36.02 \pm 2.19$ |
| SentBERT | $14.39 \pm 1.97$ | $42.42 \pm 2.45$ | $32.63 \pm 2.60$ | $69.70 \pm 1.11$ | $22.94 \pm 2.21$ | $13.28 \pm 2.71$ | $37.47 \pm 2.34$ |
| RocketQA-v2 | $10.59 \pm 1.20$ | $33.72 \pm 2.21$ | $28.42 \pm 2.18$ | $66.17 \pm 1.17$ | $18.57 \pm 2.01$ | $11.51 \pm 2.49$ | $32.57 \pm 2.14$ |
| $\text{ANCE}_{\text{FirstP}}$ | $11.10 \pm 1.53$ | $32.09 \pm 2.09$ | $28.60 \pm 2.14$ | $64.81 \pm 0.99$ | $17.18 \pm 1.74$ | $9.07 \pm 1.93$ | $30.72 \pm 2.02$ |
| SimLM | $12.68 \pm 1.77$ | $35.67 \pm 2.49$ | $28.90 \pm 2.42$ | $66.31 \pm 1.10$ | $18.91 \pm 1.86$ | $11.29 \pm 2.44$ | $33.06 \pm 2.34$ |
| SimCSE | $14.90 \pm 1.89$ | $42.62 \pm 2.40$ | $35.27 \pm 2.34$ | $70.18 \pm 1.01$ | $26.88 \pm 2.36$ | $21.19 \pm 3.47$ | $39.02 \pm 2.35$ |
| ColBERT-v2 | $8.45 \pm 1.46$ | $27.86 \pm 2.29$ | $22.33 \pm 2.01$ | $59.81 \pm 1.16$ | $12.57 \pm 1.71$ | $6.69 \pm 2.15$ | $25.80 \pm 1.83$ |
| SciBERT | $5.13 \pm 1.25$ | $17.99 \pm 1.69$ | $17.13 \pm 1.88$ | $47.28 \pm 1.17$ | $7.50 \pm 1.34$ | $3.41 \pm 1.57$ | $20.34 \pm 1.64$ |
| $\text{SciBERT}_{\text{ID}}$ | $12.86 \pm 1.86$ | $35.53 \pm 2.23$ | $27.54 \pm 2.29$ | $64.55 \pm 1.17$ | $17.24 \pm 1.64$ | $8.57 \pm 1.79$ | $31.50 \pm 2.02$ |
| ERNIE | $6.49 \pm 0.94$ | $22.58 \pm 1.72$ | $20.18 \pm 1.82$ | $54.91 \pm 1.17$ | $9.66 \pm 1.18$ | $3.77 \pm 1.06$ | $22.71 \pm 1.65$ |
| BM25 | $8.47 \pm 1.80$ | $30.50 \pm 2.38$ | $21.94 \pm 2.03$ | $51.46 \pm 1.40$ | $13.23 \pm 1.97$ | $9.19 \pm 2.46$ | $25.99 \pm 1.68$ |
| TF-IDF | $10.71 \pm 1.48$ | $29.22 \pm 2.25$ | $24.79 \pm 2.06$ | $58.31 \pm 1.12$ | $18.25 \pm 2.01$ | $12.41 \pm 2.53$ | $28.77 \pm 1.81$ |

### C.2.2 Sentence-by-Sentence Embedding

For fairness, we adopted an alternative approach for models like RocketQA-v2 [63] and ColBERT-v2 [66] that are trained on shorter queries such as in MS MARCO [57] and NQ [40]. In Table 12, we show the results when these models process each sentence in the query independently. Furthermore, we also note that TSAspire and OTAspire by their design can only encode both the query and the abstract into multiple vectors, as they utilize contextualized sentence embedding vectors for "aspects". Note, they do not process each sentence independently. These two models employ the distance function described in [54] to assess the relationship between the query vectors and the abstract vectors.

To calculate relevance between the query $Q$, which is encoded into a list of vectors $\{\vec{Q_1}, \ldots, \vec{Q_n}\}$ and the abstract $p$, which is encoded into a list of vectors $\{\vec{p_1}, \ldots, \vec{p_m}\}$, we employ the standard max-sim operation used in [35], specified by Equation 4. We report our results in Table 12.

$$\text{Rel}(p|Q) = \frac{1}{n} \sum_{i=1}^{n} \max_{1 \le j \le m} \frac{<\vec{Q_i}, \vec{p_j}>}{||\vec{Q_i}|| \cdot ||\vec{p_j}||} \qquad (4)$$

We can see RocketQA-v2 benefited the most from processing the text sentence by sentence, reaching the highest recorded performance across 6 metrics so far, which suggests a potential reason for the weakness of other NIR models, that they are unsuitable to process longer and more complex queries.

As mentioned before, in Table 2 in the main paper, we took the best recorded metrics for each model over these two options (Table 11 and Table 12).

### C.3 Using Aspects for Retrieval

In Section 5.2, we explored using the hierarchical structure of aspects and sub-aspects as an alternative to traditional queries. For this analysis, we concatenated the aspects into a paragraph to compute query embeddings for various models. However, this resulted in a verbose and repetitive paragraph, which did not resemble a real query. Given that some methods are trained specifically for sentence-level tasks, we offered the option of using a multi-vector representation for the aspects, with one vector per aspect. Similar to Sections C.2.1 and C.2.2, we considered two ways to process queries and abstracts: Approach 1 concatenates aspects and computes a single embedding; Approach 2 embeds aspects and abstracts sentence-by-sentence (this is denoted with a * in the table). For clarity, we only record the best metrics for each model between these two options in Table 13.

Table 12: Models' query-level ranking performance (embedding query and abstract into multiple vectors)

| Method | R@5 | R@20 | RP | $\text{NDCG}_{10\%}$ | $\text{NDCG}^{\text{exp}}_{10\%}$ | MRR@10 | MAP |
|---|---|---|---|---|---|---|---|
| random | 4.41 | 18.48 | 16.29 | 48.72 | 7.31 | 3.59 | 19.63 |
| SentBERT | $14.16 \pm 1.95$ | $44.81 \pm 2.54$ | $33.84 \pm 2.50$ | $69.58 \pm 0.96$ | $22.03 \pm 2.06$ | $13.39 \pm 2.63$ | $37.89 \pm 2.34$ |
| RocketQA | $\textbf{15.55} \pm 1.93$ | $\textbf{45.38} \pm 2.43$ | $\textbf{34.43} \pm 2.39$ | $\textbf{73.07} \pm 1.02$ | $\textbf{30.27} \pm 2.29$ | $\textbf{20.79} \pm 3.06$ | $\textbf{40.25} \pm 2.28$ |
| SPLADE-v2 | $12.84 \pm 1.83$ | $37.86 \pm 2.41$ | $29.30 \pm 2.40$ | $68.06 \pm 0.98$ | $23.78 \pm 2.13$ | $16.53 \pm 2.71$ | $34.67 \pm 2.16$ |
| TSAspire | $14.26 \pm 1.80$ | $41.25 \pm 2.40$ | $33.81 \pm 2.47$ | $71.63 \pm 1.11$ | $26.63 \pm 2.05$ | $15.59 \pm 2.59$ | $37.00 \pm 2.29$ |
| OTAspire | $13.34 \pm 1.56$ | $42.25 \pm 2.53$ | $33.63 \pm 2.38$ | $70.43 \pm 1.08$ | $25.52 \pm 2.29$ | $14.18 \pm 2.66$ | $36.70 \pm 2.22$ |
| ANCE | $13.17 \pm 1.89$ | $34.55 \pm 2.11$ | $30.33 \pm 2.50$ | $66.83 \pm 1.01$ | $20.27 \pm 1.91$ | $13.83 \pm 2.45$ | $34.37 \pm 2.33$ |
| SimLM | $13.77 \pm 1.87$ | $38.11 \pm 2.26$ | $30.17 \pm 2.30$ | $67.35 \pm 1.14$ | $23.61 \pm 2.14$ | $16.63 \pm 2.88$ | $34.94 \pm 2.14$ |
| ColBERT-v2 | $9.19 \pm 1.27$ | $30.28 \pm 2.03$ | $23.77 \pm 1.97$ | $61.99 \pm 1.05$ | $13.79 \pm 1.48$ | $7.20 \pm 2.13$ | $28.36 \pm 1.84$ |
| SciBERT | $5.37 \pm 0.83$ | $22.34 \pm 2.01$ | $19.44 \pm 1.86$ | $53.19 \pm 1.24$ | $8.01 \pm 1.17$ | $4.08 \pm 1.38$ | $22.54 \pm 1.62$ |
| $\text{SciBERT}_{\text{ID}}$ | $12.46 \pm 1.88$ | $35.40 \pm 2.17$ | $27.24 \pm 2.27$ | $64.67 \pm 1.18$ | $18.42 \pm 2.08$ | $12.74 \pm 2.63$ | $32.01 \pm 2.03$ |
| ERNIE | $6.71 \pm 1.39$ | $24.24 \pm 1.96$ | $21.39 \pm 2.08$ | $54.27 \pm 1.02$ | $9.92 \pm 1.50$ | $6.05 \pm 2.02$ | $24.00 \pm 1.81$ |
| SimCSE | $13.21 \pm 1.86$ | $35.83 \pm 2.31$ | $29.41 \pm 2.23$ | $65.20 \pm 1.17$ | $20.62 \pm 1.84$ | $13.59 \pm 2.58$ | $33.89 \pm 2.17$ |

When we compared Table 13 with Tables 12 and 11, we did not observe any clear effect of using aspects for retrieval. Sentence-BERT and SPLADE-v2 did achieve stronger results in this setting, but the effect did not generalize to other models.

Table 13: Models' ranking performance using concatenated aspects as query

| Method | R@5 | R@20 | RP | $\text{NDCG}_{10\%}$ | $\text{NDCG}^{\text{exp}}_{10\%}$ | MRR@10 | MAP |
|---|---|---|---|---|---|---|---|
| random | 4.60 | 18.30 | 16.48 | 48.79 | 7.41 | 3.65 | 19.73 |
| SentBERT* | $\textbf{17.71} \pm 2.44$ | $\textbf{45.26} \pm 2.66$ | $\textbf{35.63} \pm 2.48$ | $70.47 \pm 0.97$ | $24.94 \pm 2.19$ | $15.54 \pm 3.00$ | $\textbf{39.79} \pm 2.42$ |
| RocketQA* | $13.79 \pm 1.48$ | $43.69 \pm 2.38$ | $32.63 \pm 2.17$ | $\textbf{72.21} \pm 0.93$ | $\textbf{27.39} \pm 2.04$ | $16.03 \pm 2.52$ | $37.90 \pm 2.13$ |
| ANCE* | $15.35 \pm 2.02$ | $38.36 \pm 2.29$ | $29.45 \pm 2.07$ | $67.27 \pm 1.08$ | $20.46 \pm 1.82$ | $14.54 \pm 2.75$ | $33.59 \pm 1.89$ |
| SPLADE-v2 | $15.07 \pm 1.99$ | $41.20 \pm 2.42$ | $34.26 \pm 2.45$ | $71.34 \pm 0.91$ | $24.91 \pm 1.90$ | $18.30 \pm 3.00$ | $38.72 \pm 2.33$ |
| ColBERT-v2* | $7.37 \pm 1.11$ | $26.64 \pm 1.94$ | $22.28 \pm 1.82$ | $58.70 \pm 1.14$ | $12.17 \pm 1.30$ | $5.78 \pm 1.65$ | $25.24 \pm 1.69$ |
| ERNIE* | $5.34 \pm 1.23$ | $19.96 \pm 1.58$ | $18.02 \pm 1.64$ | $50.33 \pm 1.13$ | $7.14 \pm 1.03$ | $3.14 \pm 1.26$ | $21.38 \pm 1.50$ |
| SciBERT* | $5.41 \pm 0.78$ | $21.36 \pm 1.86$ | $18.50 \pm 1.72$ | $51.27 \pm 1.15$ | $6.84 \pm 0.92$ | $2.29 \pm 1.04$ | $21.66 \pm 1.57$ |
| $\text{SciBERT}_{\text{ID}}$* | $8.88 \pm 1.44$ | $30.81 \pm 2.15$ | $22.56 \pm 2.00$ | $59.86 \pm 1.01$ | $13.82 \pm 1.64$ | $10.24 \pm 2.28$ | $26.53 \pm 1.77$ |
| ada-002 | $14.12 \pm 2.00$ | $42.13 \pm 2.73$ | $33.60 \pm 2.44$ | $71.52 \pm 0.98$ | $26.57 \pm 2.57$ | $\textbf{20.22} \pm 3.49$ | $37.63 \pm 2.29$ |
| E5-L-v2 | $14.32 \pm 1.97$ | $41.55 \pm 2.21$ | $34.21 \pm 2.54$ | $69.37 \pm 1.00$ | $22.51 \pm 2.16$ | $13.87 \pm 2.68$ | $37.31 \pm 2.33$ |
| TSAspire | $14.30 \pm 1.80$ | $42.75 \pm 2.25$ | $33.63 \pm 2.45$ | $70.96 \pm 1.10$ | $25.29 \pm 2.14$ | $16.26 \pm 2.90$ | $37.31 \pm 2.23$ |
| OTAspire | $14.45 \pm 1.81$ | $40.99 \pm 2.30$ | $32.53 \pm 2.43$ | $69.74 \pm 1.14$ | $23.81 \pm 2.07$ | $15.38 \pm 2.76$ | $36.09 \pm 2.18$ |
| SimLM | $16.18 \pm 2.19$ | $39.24 \pm 2.36$ | $32.20 \pm 2.22$ | $67.56 \pm 1.03$ | $23.36 \pm 2.14$ | $16.61 \pm 2.84$ | $36.24 \pm 2.12$ |
| SimCSE | $12.41 \pm 1.84$ | $35.02 \pm 2.52$ | $27.67 \pm 2.27$ | $64.19 \pm 1.24$ | $20.52 \pm 1.93$ | $15.27 \pm 2.57$ | $30.83 \pm 2.12$ |
| SPECTER | $11.56 \pm 1.76$ | $35.30 \pm 2.34$ | $28.43 \pm 2.49$ | $63.55 \pm 1.30$ | $19.69 \pm 2.33$ | $15.07 \pm 3.12$ | $31.39 \pm 2.22$ |
| $\text{SPECTER}_{\text{ID}}$ | $12.01 \pm 1.77$ | $35.79 \pm 2.27$ | $28.71 \pm 2.17$ | $66.22 \pm 1.05$ | $19.70 \pm 2.04$ | $13.44 \pm 2.67$ | $32.69 \pm 1.97$ |
| SPECTER-v2 | $12.97 \pm 1.78$ | $41.45 \pm 2.50$ | $31.30 \pm 2.35$ | $68.37 \pm 1.13$ | $23.16 \pm 2.20$ | $14.59 \pm 2.66$ | $35.77 \pm 2.09$ |
| BM25 | $6.25 \pm 1.00$ | $25.11 \pm 2.09$ | $20.51 \pm 1.88$ | $49.65 \pm 1.31$ | $10.33 \pm 1.33$ | $5.72 \pm 1.62$ | $23.74 \pm 1.57$ |
| TF-IDF | $7.62 \pm 1.40$ | $23.16 \pm 2.10$ | $20.14 \pm 2.09$ | $50.73 \pm 1.52$ | $12.55 \pm 1.81$ | $7.72 \pm 2.15$ | $24.25 \pm 1.91$ |

* : Approach 2

## C.4 Model Performance on DORIS-MAE Test Set

In Appendix section C.2, we evaluated zero-shot performance of embedding models on the full DORIS-MAE dataset. We now present zero-shot results on the DORIS-MAE test set, which consists of 60 queries. For these queries, we run all IR models and show their performances in Table 14. All models use their individual optimal embedding method (i.e. processing query/abstract sentence by sentence or as an entire paragraph). Results from this table serve as a baseline for future work that trains models on the DORIS-MAE training set.

Table 14: Models' query-level ranking performance on DORIS-MAE test set

| Method | R@5 | R@20 | RP | NDCG$_{10\%}$ | NDCG$_{10\%}^{\text{exp}}$ | MRR@10 | MAP |
|---|---|---|---|---|---|---|---|
| random | 4.58 | 17.91 | 17.34 | 50.22 | 7.36 | 3.32 | 20.69 |
| ada-002 | **15.80** $\pm$ 1.92 | 40.87 $\pm$ 3.09 | 37.64 $\pm$ 3.17 | **73.92** $\pm$ 1.29 | 28.47 $\pm$ 3.22 | 23.29 $\pm$ 4.52 | **41.72** $\pm$ 3.12 |
| SimCSE | 14.40 $\pm$ 1.71 | 41.94 $\pm$ 3.01 | 36.29 $\pm$ 2.86 | 71.31 $\pm$ 1.25 | 29.08 $\pm$ 3.16 | **25.00** $\pm$ 4.85 | 39.65 $\pm$ 2.86 |
| RocketQA | 13.26 $\pm$ 1.29 | **44.35** $\pm$ 2.80 | 35.32 $\pm$ 2.97 | 73.87 $\pm$ 1.26 | **29.83** $\pm$ 2.88 | 20.61 $\pm$ 4.01 | 40.66 $\pm$ 2.84 |
| SPECTER-v2 | 13.33 $\pm$ 2.26 | 41.83 $\pm$ 3.07 | 35.98 $\pm$ 3.16 | 71.46 $\pm$ 1.15 | 27.67 $\pm$ 3.26 | 20.70 $\pm$ 4.47 | 38.46 $\pm$ 2.80 |
| TSAspire | 13.31 $\pm$ 1.67 | 43.57 $\pm$ 2.97 | 36.51 $\pm$ 2.96 | 72.78 $\pm$ 1.26 | 27.72 $\pm$ 2.81 | 18.22 $\pm$ 3.81 | 38.50 $\pm$ 2.87 |
| E5-L-v2 | 14.59 $\pm$ 1.79 | 42.06 $\pm$ 2.47 | **38.16** $\pm$ 2.93 | 71.96 $\pm$ 1.18 | 26.04 $\pm$ 2.76 | 14.25 $\pm$ 3.52 | 40.51 $\pm$ 2.87 |
| OTAspire | 13.12 $\pm$ 1.61 | 42.65 $\pm$ 3.22 | 35.35 $\pm$ 2.97 | 71.01 $\pm$ 1.36 | 25.69 $\pm$ 2.72 | 15.38 $\pm$ 3.22 | 37.83 $\pm$ 2.86 |
| SPLADE-v2 | 10.99 $\pm$ 1.18 | 36.21 $\pm$ 2.57 | 32.50 $\pm$ 2.96 | 71.01 $\pm$ 1.18 | 25.10 $\pm$ 2.90 | 19.44 $\pm$ 4.38 | 36.44 $\pm$ 2.81 |
| SentBERT | 12.18 $\pm$ 1.72 | 43.23 $\pm$ 2.98 | 34.38 $\pm$ 3.07 | 69.63 $\pm$ 1.31 | 19.37 $\pm$ 2.47 | 10.84 $\pm$ 3.34 | 37.72 $\pm$ 2.93 |
| SimLM | 10.49 $\pm$ 1.27 | 32.19 $\pm$ 2.72 | 29.73 $\pm$ 3.14 | 66.73 $\pm$ 1.36 | 17.50 $\pm$ 2.52 | 11.90 $\pm$ 3.66 | 32.28 $\pm$ 3.04 |
| ANCE$_{\text{FirstP}}$ | 10.88 $\pm$ 1.83 | 32.03 $\pm$ 2.72 | 29.02 $\pm$ 3.04 | 66.48 $\pm$ 1.32 | 17.92 $\pm$ 2.28 | 11.11 $\pm$ 2.79 | 32.90 $\pm$ 2.83 |
| LLAMA | 9.60 $\pm$ 1.36 | 32.34 $\pm$ 2.86 | 28.43 $\pm$ 2.69 | 65.15 $\pm$ 1.24 | 14.71 $\pm$ 1.84 | 9.36 $\pm$ 2.67 | 31.16 $\pm$ 2.60 |
| TF-IDF | 9.53 $\pm$ 1.42 | 26.77 $\pm$ 2.51 | 25.27 $\pm$ 2.50 | 58.14 $\pm$ 1.59 | 17.03 $\pm$ 2.39 | 11.59 $\pm$ 3.36 | 28.56 $\pm$ 2.28 |
| BM25 | 6.86 $\pm$ 1.52 | 26.43 $\pm$ 2.27 | 22.45 $\pm$ 2.51 | 51.55 $\pm$ 1.76 | 12.29 $\pm$ 2.55 | 8.65 $\pm$ 3.49 | 26.00 $\pm$ 2.16 |
| ColBERTv2 | 6.49 $\pm$ 1.21 | 23.99 $\pm$ 2.78 | 21.58 $\pm$ 2.66 | 59.46 $\pm$ 1.64 | 12.92 $\pm$ 2.49 | 7.15 $\pm$ 2.97 | 25.24 $\pm$ 2.41 |
| ERNIE | 7.43 $\pm$ 1.26 | 24.92 $\pm$ 2.23 | 22.43 $\pm$ 2.50 | 57.52 $\pm$ 1.50 | 10.83 $\pm$ 1.62 | 3.64 $\pm$ 1.28 | 24.94 $\pm$ 2.24 |
| SciBERT | 4.57 $\pm$ 1.08 | 17.35 $\pm$ 1.84 | 16.83 $\pm$ 2.27 | 49.42 $\pm$ 1.56 | 7.13 $\pm$ 1.19 | 2.00 $\pm$ 1.06 | 20.61 $\pm$ 1.91 |

## C.5 Two-Aspect Sub-Query Level Benchmark Results

The dataset's hierarchical structure provides an opportunity to formulate simpler tasks involving only parts of a complex query. By extracting parts corresponding to two aspects, we generated over 1000 test cases, significantly increasing the number of relevant abstracts as outlined in Section A.5. The full results for this task are given in Table 15. As the length of these sub-queries is similar to a long sentence, we did not differentiate between embedding the entire query as a single vector or multiple vectors sentence-by-sentence. Except for OTAspire and TSAspire, we applied a single-vector embedding for all other models, including Sentence-BERT and ANCE.

We observed that ada-002 outperformed other models across all categories. All models exhibited slight increases in R-Precision, NDCG$_{10\%}^{\text{exp}}$, and MAP. These minor improvements correlate with similar increases in their random baselines, suggesting that they might be due to shifts in the random baseline rather than the models' retrieval capabilities.Meanwhile, the small decreases in the random baselines of R@5 and R@20 indicate a drop in the retrieval models' Recall@k performance on the sub-query level. This corresponds to an increase in the number of relevant abstracts per sub-query as shown in Table 9. As Recall@k is inversely related to the number of relevant abstracts per sub-query, and as the retrieval models do not appear to effectively retrieve relevant abstracts for these simpler sub-queries, it is unsurprising that their R@5 and R@20 metrics dropped substantially.

## C.6 Candidate Pool Sensitivity Analyses

In this section, we perform a sensitivity analysis on the candidate pools in DORIS-MAE. The original candidate pools were primarily constructed using keyword-based methods. To evaluate how this may have influenced our results, we generated a second set of candidate pools for the first 50 queries in DORIS-MAE. These new "embedding-based candidate pools" were generated with two text-embedding models, E5-v2 and SPECTER-v2, which were not used during the original candidate pool creation.

For each query, we collect the top-ranking 150 abstracts that are not in the query's original candidate pool (75 from E5-v2 and 75 from SPECTER-v2). We applied the same LLM annotation procedure to estimate the relevance of each abstract in the candidate pool given the query.

**Relevance Analysis:** We first evaluated whether the new pool contains documents which are more relevant than those in the original pool. For each of the 50 queries, and for each candidate pool (i.e. the original pool and the extended embedding pool), we record the highest relevance score. For each query, we then compare the highest relevance score from each pooling method. On average, the

Table 15: Models' ranking performance on 2-aspect sub-queries

| Method | R@5 | R@20 | RP | NDCG$_{10\%}$ | NDCG$_{10\%}^{\text{exp}}$ | MRR@10 | MAP |
|---|---|---|---|---|---|---|---|
| random | 4.60 | 18.76 | 21.55 | 41.10 | 15.06 | 5.92 | 24.71 |
| ada-002 | **13.47** ± 0.47 | 40.22 ± 0.64 | 47.34 ± 0.77 | **69.78** ± 0.39 | **38.97** ± 0.64 | **24.29** ± 1.07 | **51.65** ± 0.77 |
| LLAMA | 8.06 ± 0.31 | 29.77 ± 0.61 | 36.49 ± 0.69 | 56.85 ± 0.44 | 23.59 ± 0.50 | 9.80 ± 0.66 | 39.01 ± 0.65 |
| E5-L-v2 | 11.62 ± 0.37 | 38.26 ± 0.58 | 45.11 ± 0.78 | 65.82 ± 0.42 | 35.12 ± 0.59 | 21.00 ± 0.95 | 48.73 ± 0.76 |
| TSAspire | 13.15 ± 0.69 | 43.47 ± 1.04 | 41.46 ± 1.24 | 65.54 ± 0.71 | 36.31 ± 0.97 | 23.54 ± 1.58 | 45.85 ± 1.20 |
| OTAspire | 12.56 ± 0.70 | **43.66** ± 1.09 | 40.60 ± 1.23 | 65.06 ± 0.69 | 35.54 ± 0.91 | 22.71 ± 1.49 | 45.53 ± 1.16 |
| SPECTER | 10.51 ± 0.85 | 35.99 ± 1.07 | 35.40 ± 1.13 | 55.71 ± 0.67 | 26.26 ± 0.95 | 14.06 ± 1.32 | 38.27 ± 1.05 |
| SPECTER-v2 | 11.14 ± 0.41 | 36.09 ± 0.60 | 43.73 ± 0.80 | 64.80 ± 0.41 | 32.68 ± 0.55 | 18.02 ± 0.90 | 47.26 ± 0.79 |
| SPECTER$_{\text{ID}}$ | 8.05 ± 0.31 | 30.97 ± 0.56 | 38.11 ± 0.77 | 57.19 ± 0.43 | 23.96 ± 0.45 | 10.09 ± 0.64 | 40.23 ± 0.74 |
| SentBERT | 12.79 ± 0.42 | 40.31 ± 0.62 | **48.34** ± 0.80 | 69.38 ± 0.42 | 38.52 ± 0.61 | 22.98 ± 1.00 | 52.34 ± 0.80 |
| RocketQA | 13.37 ± 0.61 | 42.42 ± 1.00 | 42.65 ± 1.19 | 67.20 ± 0.70 | 37.33 ± 1.04 | 24.19 ± 1.65 | 46.39 ± 1.21 |
| ANCE$_{\text{FirstP}}$ | 11.23 ± 0.43 | 35.12 ± 0.58 | 42.49 ± 0.77 | 64.00 ± 0.43 | 31.64 ± 0.62 | 17.70 ± 0.96 | 45.78 ± 0.77 |
| SimLM | 12.61 ± 0.70 | 37.73 ± 1.02 | 37.90 ± 1.08 | 63.18 ± 0.69 | 31.59 ± 0.95 | 19.18 ± 1.46 | 41.82 ± 1.10 |
| SPLADE-v2 | 12.95 ± 0.46 | 39.55 ± 0.62 | 46.63 ± 0.75 | 68.26 ± 0.39 | 37.33 ± 0.60 | 22.80 ± 1.00 | 50.71 ± 0.75 |
| ColBERT-v2 | 8.13 ± 0.50 | 27.22 ± 0.87 | 28.38 ± 1.02 | 53.32 ± 0.71 | 22.72 ± 0.79 | 11.84 ± 1.19 | 32.03 ± 0.98 |
| ERNIE | 3.25 ± 0.21 | 15.96 ± 0.53 | 20.05 ± 0.87 | 39.26 ± 0.69 | 12.83 ± 0.56 | 3.29 ± 0.66 | 23.27 ± 0.79 |
| SciBERT | 2.43 ± 0.20 | 14.59 ± 0.71 | 17.56 ± 0.78 | 33.29 ± 0.58 | 9.65 ± 0.42 | 1.35 ± 0.24 | 21.23 ± 0.72 |
| SciBERT$_{\text{ID}}$ | 9.41 ± 0.35 | 32.43 ± 0.56 | 39.28 ± 0.72 | 60.19 ± 0.42 | 27.91 ± 0.51 | 13.63 ± 0.80 | 42.19 ± 0.70 |
| BM25 | 9.14 ± 0.62 | 31.68 ± 0.99 | 31.22 ± 1.00 | 51.05 ± 0.71 | 25.52 ± 0.84 | 14.74 ± 1.18 | 33.94 ± 0.94 |
| TF-IDF | 9.69 ± 0.58 | 32.56 ± 0.93 | 33.32 ± 1.05 | 56.58 ± 0.67 | 28.79 ± 0.91 | 19.57 ± 1.52 | 36.41 ± 0.97 |
| SimCSE | 10.98 ± 0.38 | 34.57 ± 0.58 | 41.40 ± 0.74 | 63.00 ± 0.43 | 31.19 ± 0.56 | 17.22 ± 0.90 | 44.96 ± 0.74 |

most relevant documents in the original pool scored higher than those in the embedding pool by 4%. However, for 30% of the queries, the embedding pool did contain a more relevant document, with an average improvement of 12% over the original pool. The results indicate that the best documents in the original candidate pool were comparable to those in the extended embedding pool.

**Reranking Performance Analysis:** Next, we compared the reranking performance of the IR models on the two pools. Comparing Table 16 with Table 2 and Table 14, we noticed significant performance decreases in E5-v2, SPECTER-v2, SimCSE and SentBERT on the new pool. The performance of ada-002 remains stable. Since all abstracts in the new extended candidate pool are considered highly relevant by the pretrained embeddings of E5-v2 and SPECTER-v2, which are the methods used for the extended candidate pool generation, the performance drops for these two models can be attributed to selection bias that makes it harder for the same reranking models to distinguish among their own pre-selected documents. Furthermore, transformer-based encoder models may be generating similar document representations, since most of these models also see performance drops. The results suggest that it is useful to generate candidate pools using methods that have low correlation with downstream reranking methods.

# D   Model Details

## D.1   Retraining Details

We retrained SPECTER and SciBERT on our corpus of 360k CS related papers with citation information. In particular, we retrained SPECTER for 4 epochs, but noticed decreasing performance beyond 1 epoch. Therefore, only the epoch 1 checkpoint is kept for SPECTER$_{\text{ID}}$. We leveraged the citation signals from Semantic Scholar [37] to train the SPECTER model. Abstract A is related to abstract B if A directly cites B or if A cites a paper that cites B. Our corpus consists of 282,121 abstracts with related abstracts by the above definition. Among these abstracts, each has an average of 76.327 related abstracts. Consequently, we utilize a total of 21,533,311 triplets during the training of SPECTER. We retrained SciBERT for 10 epochs using 8 GPUs in 12 hours. Checkpoint 10 is kept for SciBERT$_{\text{ID}}$. Figure 3 plots the training loss curve for SPECTER and SciBERT.

Table 16: Models' query-level ranking performance on the extended 150-abstracts candidate pool

| Method | R@5 | R@20 | RP | NDCG$_{10\%}$ | NDCG$_{10\%}^{exp}$ | MRR@10 | MAP |
|---|---|---|---|---|---|---|---|
| random | 3.28 | 13.47 | 9.36 | 47.73 | 6.71 | 3.25 | 12.37 |
| ada-002 | **15.39** ± 3.49 | **37.17** ± 3.86 | **26.64** ± 3.52 | **67.45** ± 1.42 | **23.66** ± 3.34 | **20.97** ± 5.20 | **29.66** ± 3.35 |
| RocketQA | 11.33 ± 1.70 | 36.04 ± 2.57 | 21.51 ± 2.77 | 68.42 ± 1.40 | 21.40 ± 2.92 | 14.12 ± 3.57 | 24.62 ± 2.36 |
| SimCSE | 12.87 ± 3.30 | 35.92 ± 4.11 | 18.39 ± 2.37 | 64.22 ± 1.28 | 19.24 ± 1.95 | 8.34 ± 2.04 | 22.16 ± 1.82 |
| SentBERT | 6.55 ± 1.51 | 31.21 ± 3.44 | 18.41 ± 2.65 | 64.21 ± 1.42 | 16.93 ± 2.50 | 10.06 ± 3.41 | 22.50 ± 2.28 |
| TSAspire | 6.17 ± 1.24 | 28.43 ± 3.36 | 17.42 ± 2.36 | 62.08 ± 1.20 | 13.58 ± 1.74 | 4.90 ± 1.39 | 19.54 ± 1.92 |
| OTAspire | 6.12 ± 1.21 | 26.44 ± 2.95 | 17.28 ± 2.31 | 62.34 ± 1.21 | 13.83 ± 1.76 | 5.62 ± 1.79 | 19.35 ± 2.03 |
| SPLADE-v2 | 10.04 ± 3.14 | 28.36 ± 3.45 | 14.05 ± 2.24 | 59.09 ± 1.43 | 12.10 ± 1.89 | 5.75 ± 1.95 | 18.48 ± 1.76 |
| ANCE$_{FirstP}$ | 9.69 ± 2.61 | 24.31 ± 2.77 | 17.87 ± 2.81 | 61.10 ± 1.27 | 17.04 ± 2.99 | 11.50 ± 3.70 | 20.70 ± 2.60 |
| SPECTER-v2 | 6.75 ± 1.19 | 20.17 ± 2.03 | 15.18 ± 1.98 | 59.11 ± 1.46 | 12.54 ± 1.93 | 8.45 ± 3.15 | 17.37 ± 1.83 |
| SimLM | 5.55 ± 1.04 | 24.88 ± 2.69 | 16.22 ± 2.08 | 60.54 ± 1.23 | 13.27 ± 1.74 | 4.76 ± 2.23 | 18.11 ± 1.66 |
| TF-IDF | 5.22 ± 1.55 | 19.64 ± 2.71 | 14.15 ± 2.24 | 55.93 ± 1.38 | 15.32 ± 2.97 | 9.58 ± 3.59 | 17.24 ± 1.90 |
| E5-L-v2 | 4.88 ± 0.93 | 13.01 ± 1.99 | 11.43 ± 2.00 | 48.53 ± 1.80 | 9.04 ± 1.59 | 2.92 ± 1.24 | 14.48 ± 1.90 |
| ColBERT-v2 | 6.73 ± 2.35 | 17.40 ± 2.58 | 11.58 ± 1.72 | 57.10 ± 1.34 | 8.22 ± 1.21 | 2.14 ± 0.84 | 14.70 ± 1.41 |
| ERNIE | 4.25 ± 1.22 | 13.86 ± 2.57 | 10.39 ± 1.64 | 47.05 ± 1.65 | 6.91 ± 1.40 | 2.78 ± 1.42 | 12.89 ± 1.42 |
| BM25 | 2.16 ± 0.68 | 21.13 ± 2.81 | 10.36 ± 1.66 | 47.76 ± 1.20 | 8.26 ± 1.57 | 2.69 ± 1.17 | 13.58 ± 1.35 |
| SciBERT | 0.69 ± 0.29 | 5.65 ± 1.36 | 4.60 ± 1.05 | 34.73 ± 1.30 | 3.00 ± 0.86 | 1.17 ± 0.65 | 8.91 ± 1.03 |

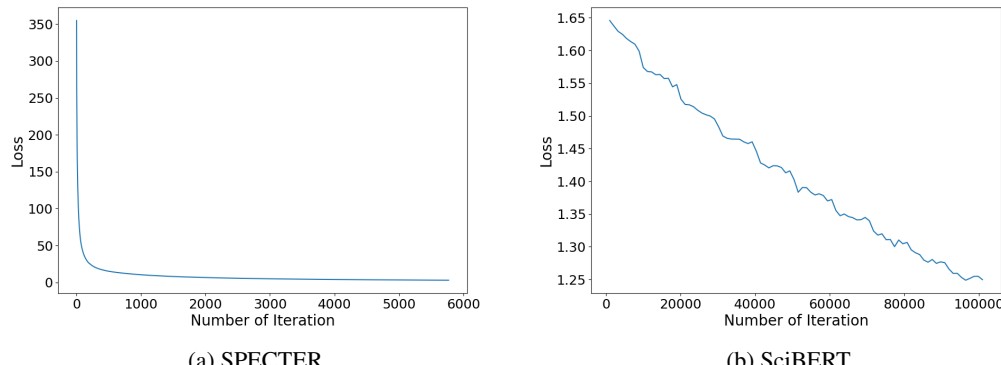

(a) SPECTER               (b) SciBERT

Figure 3: Retrained Models

## D.2 Model Specifications

### D.2.1 Sentence-level Embedding Models

Both ANCE [85] and SentBERT [62] are trained for sentence-level tasks. Instead of using these models to embed the entire query and abstract, we use them to embed individual sentences of query and abstract.

Approximate nearest neighbor Negative Contrastive Estimation(ANCE) FirstP is a BERT-based model that conducts text retrieval in dense multi-vector representations. The model optimizes the representation of data during training by selecting challenging negative samples using an Approximate Nearest Neighbor (ANN) search. The FirstP model uses the first 512 tokens of the document. The method's implementation comes from https://huggingface.co/sentence-transformers/msmarco-roberta-base-ance-firstp.

Sentence-BERT is an enhancement of the traditional BERT-based transformer model optimized for sentence-level similarity tasks. It uses Siamese and triplet network structures to encode documents into semantically meaningful sentence embeddings. The model is trained on Stanford Natural Language Inference (SNLI) corpus [7] and the Multi-Genre Natural Language Inference (MultiNLI) corpus [84]. The method's implementation comes from https://www.sbert.net/#.

### D.2.2 Passage Retrieval Models

ColBERT-v2 [66], ERNIE-Search [47], SimLM [75], SPLADE-v2 [23], and RocketQA-v2 [63] are models specifically engineered for passage retrieval tasks.

ColBERT-v2 is a retrieval model that incorporates lightweight late interactions on dense multi-vector representations. The BERT-based ColBERT-v2 model is trained on MS MARCO [57]. By incorporating an aggressive residual compression mechanism and a denoised supervision strategy, ColBERT-v2 achieves more efficient retrieval while maintaining high-quality results. The method's implementation comes from https://github.com/stanford-futuredata/ColBERT.

ERNIE-Search is a neural retrieval model optimized for open-domain question answering tasks. Built on a pre-trained language model, ERNIE-Search leverages a dual-encoder architecture and introduces a novel method of cross-architecture distillation to improve its performance. The model is trained on MS MARCO and NQ [57, 40]. The model's implementation comes from https://huggingface.co/docs/transformers/model_doc/ernie.

Similarity matching with Language Model pre-training (SimLM), is a retrieval model that embeds queries and abstracts into single-vector representations. It employs a simple bottleneck architecture that encodes documents into a dense vector through self-supervised pre-training. SimLM model is fine-tuned on MS MARCO passage corpus. The method's implementation comes from https://github.com/microsoft/unilm/tree/master/simlm.

Sparse Lexical and Expansion Model for First Stage Ranking (SPLADE-v2) is a model that offers highly sparse representations for documents and queries for information retrieval tasks. The model is trained on MS MARCO passage ranking dataset. The method's implementation follows the implementation in BEIR evaluation which comes from https://github.com/beir-cellar/beir/blob/main/beir/retrieval/models/splade.py.

RocketQA-v2 is a model that incorporates dynamic list-wise distillation mechanism for jointly training the retriever and the re-ranker. The model is trained on MS MARCO and NQ. The model's method takes a query and abstracts content as input and outputs their matching scores for the reranking process. The method's implementation comes from https://github.com/PaddlePaddle/RocketQA.

### D.2.3 Document Similarity Models

SPECTER [18] and Aspire [54] are models designed for document-level similarities.

Scientific Paper Embeddings using Citation-informed TransformER (SPECTER), is a transformer-based model that is fine-tuned on scientific documents. It generates document-level embedding of scientific documents based on the pretrained SciBERT [5] on citation graph. SPECTER was fine-tuned on a subset of the Semantic Scholar corpus. SPECTER 2.0, pre-trained on a collection of newer papers published after 2018, is the successor to SPECTER and is capable of generating task specific embeddings for scientific tasks when paired with adapters. In our experiment, we use the default adaptor. It should be noted that SPECTER uses L2 distance between vectors instead of cosine-similarity.

The method's implementation comes from https://github.com/allenai/specter/tree/master and https://huggingface.co/allenai/specter2.

Aspire [54] is a document similarity model that flexibly aggregates over fine-grained sentence-level aspect matches. It embeds the query and abstract as multiple vectors. Though each vector represents an individual sentence in the text, it is dependent on its surrounding context. The query and abstract are both embedded once to get the multi-vector representation. The similarity between query and abstract is calculated using the text-supervised single match (TSAspire) method and the optimal-transport multi-match (OTAspire) method, described in [54]. The method's implementation comes from https://github.com/allenai/aspire.

### D.2.4 General Text Embedding Models

We also use ada-002 [28], E5-Large-V2 [74] and LLAMA [71] are general text embedding models. These models are not restricted by the size of the context window and can embed longer texts. We only test these models on the full-query DORIS-MAE task C.2.1.

Ada-002 (ada) is a text embedding model provided by OpenAI. Ada leverages a transformer-based architecture and incorporates several novel techniques to enhance its embedding quality. The model employs noise reduction techniques to improve the reliability and consistency of the generated embeddings. We use ada-002 via OpenAI's API.

E5-Large-V2 300m is a text embedding model developed by Weakly-Supervised Contrastive Pre-training [74]. The model has demonstrated strong results on benchmarks such as BEIR [69]. The implementation comes from huggingface at: https://huggingface.co/intfloat/e5-large-v2.

Large Language Model Meta AI (LLAMA), is state-of-the-art foundational LLM released by Meta. The 7 billion model has been trained on about one trillion tokens. We applied for the weights of the 7 billion parameter version. We use the LLAMA to generate the text embedding by extracting the outputs from the penultimate layer of the model.

# E    Author Statement

The authors hereby affirm that we are the sole authors of the submitted manuscript and retain full responsibility for its contents. The authors warrant that this work is original, has not been published elsewhere, and is currently not under consideration for publication elsewhere. In the event of any violation of legal rights, the authors fully accept and bear all the repercussions and consequences. Furthermore, the authors confirm that all data used in this research complies with the necessary licensing requirements. The authors acknowledge that non-compliance with these statements may lead to the retraction of our work and possible legal consequences.

# F    Datasheet For DORIS-MAE

This document is based on *Datasheets for Datasets* by Gebru *et al.* [26]. Please see the most updated version here.

All questions that are not applicable or have a negative answer are omitted for brevity.

## F.1    MOTIVATION

**Q: For what purpose was the dataset created?**

A: The dataset was created for the purpose of evaluating and improving the performance of scientific document retrieval systems on complex, multi-intent user queries. This work aims to fill the gap in existing resources that often fail to capture the complexity and multifaceted nature of queries typical in a scientific research context.

. . . . . . . . . . . . . . . . . . . . . . . . . . . . . . . . . . . . . . . . . . . . . . . . . . . . . . . . . . . . . . . . . . . . . . . . . . . . . . . . . . . . . . . .

**Q: Who created this dataset (e.g., which team, research group) and on behalf of which entity (e.g., company, institution, organization)?**

A: Authors will remain anonymous until the acceptance of our work.

. . . . . . . . . . . . . . . . . . . . . . . . . . . . . . . . . . . . . . . . . . . . . . . . . . . . . . . . . . . . . . . . . . . . . . . . . . . . . . . . . . . . . . . .

**Q: What support was needed to make this dataset?**

A: Funders will remain anonymous until the acceptance of our work.

. . . . . . . . . . . . . . . . . . . . . . . . . . . . . . . . . . . . . . . . . . . . . . . . . . . . . . . . . . . . . . . . . . . . . . . . . . . . . . . . . . . . . . . .

## F.2    COMPOSITION

**Q: What do the instances that comprise the dataset represent (e.g., documents, photos, people, countries)?**

A: The dataset include several types of data instances.

- Multifaceted complex queries in text, created by human experts.
- Hierarchical lists of aspects and sub-aspects for each complex query, created by human experts.
- Candidate pools of potentially relevant paper abstracts from arXiv for each complex query.
- Annotations (human-made and ChatGPT-made) for each aspect-abstract question pair.

. . . . . . . . . . . . . . . . . . . . . . . . . . . . . . . . . . . . . . . . . . . . . . . . . . . . . . . . . . . . . . . . . . . . . . . . . . . . . . . . . . . . . . . .

**Q: How many instances are there in total (of each type, if appropriate)?**

A: It has 50 complex queries containing between 95 to 217 words, 50 Hierarchical lists of aspects and sub-aspects, 50 Candidate pools, 83,591 ChatGPT-made annotations and 250 human-made annotations.

. . . . . . . . . . . . . . . . . . . . . . . . . . . . . . . . . . . . . . . . . . . . . . . . . . . . . . . . . . . . . . . . . . . . . . . . . . . . . . . . . . . . . . . .

**Q: Does the dataset contain all possible instances or is it a sample (not necessarily random) of instances from a larger set?**

A: The dataset contains 50 queries in the fields of ML, CV, AI, and NLP. These queries are manually written. As described in Section 3.1, queries are formulated by first randomly sampling an abstract from the CS corpus, which contains 360k papers from 2011-2021 posted on arXiv, and designing queries based on these abstracts. Each candidate pool is selected from the CS corpus, which should represent close to the entirety of CS research papers during that time period.

. . . . . . . . . . . . . . . . . . . . . . . . . . . . . . . . . . . . . . . . . . . . . . . . . . . . . . . . . . . . . . . . . . . . . . . . . . . . . . . . . . . . . . . .

**Q: What data does each instance consist of?**

A: See Section A.4 for a detailed breakdown of each data instance.

......................................................................................

**Q: Is there a label or target associated with each instance?**

A: For each aspect-abstract pair, an annotation score from 0-2 is provided.

......................................................................................

**Q: Are there recommended data splits (e.g., training, development/validation, testing)?**

A: The entire DORIS-MAE dataset consists of 83,591 annotated question pairs of (aspect/sub-aspect and abstract). In particular, the dataset contains a development set of 90 manually annotated pairs for prompt optimization, a test set of 250 manually annotated pairs for hypothesis testing, and 83,591 pairs to compute the relevance rankings of abstracts for each candidate pool. These relevance rankings serve as a test benchmark for evaluating various retrieval models.

......................................................................................

**Q: Are there any errors, sources of noise, or redundancies in the dataset?**

A: The annotation of the aspect-abstract pair may contain some noise caused by GPU non-determinism. Though we have verified that ChatGPT's performance is on-par with humans', we have documented the possible inconsistencies in Section B.4.

......................................................................................

**Q: Is the dataset self-contained, or does it link to or otherwise rely on external resources (e.g., websites, tweets, other datasets)?**

A: The dataset is self-contained.

......................................................................................

### F.3 COLLECTION

**Q: How was the data associated with each instance acquired?**

A: The abstracts in the candidate pools and the corpus are from arXiv[4]. The queries/aspects/sub-aspects are manually created. The annotations are generated by chatgpt-3.5-turbo-0301.

......................................................................................

**Q: Over what timeframe was the data collected?**

A: The abstracts in the candidate pools and the corpus are from papers from 2011 to 2021. The queries are created by the authors in 2023.

......................................................................................

**Q: What mechanisms or procedures were used to collect the data?**

A: The queries/aspects/sub-aspects are manually created. The abstracts in the candidate pools/corpus are collected from the arXiv API. The citation signals of the papers are collected from Semantic Scholar API[37]. The annotations are generated by chatgpt-3.5-0301 via OpenAI API. The retireval models are retrained using 8 NVIDIA GeForce RTX 2080 Titan GPUs.

......................................................................................

**Q: What was the resource cost of collecting the data?**

A: All costs are detailed in Section 4.4 and Section B.5. Overall, the API costs less than $100.

......................................................................................

**Q: If the dataset is a sample from a larger set, what was the sampling strategy?**

---

[4]https://arxiv.org/

A: The candidate pools are sampled from the corpus by a variety of IR and NIR methods. The queries are created by human experts from existing paper abstracts, which are randomly sampled from the corpus.

. . . . . . . . . . . . . . . . . . . . . . . . . . . . . . . . . . . . . . . . . . . . . . . . . . . . . . . . . . . . . . . . . . . . . . . . . . . . . . . . . . . . . . . . . . . . . . .

**Q: Who was involved in the data collection process (e.g., students, crowdworkers, contractors) and how were they compensated (e.g., how much were crowdworkers paid)?**

A: Only the authors are involved in the data annotation process and compensation is NA.

. . . . . . . . . . . . . . . . . . . . . . . . . . . . . . . . . . . . . . . . . . . . . . . . . . . . . . . . . . . . . . . . . . . . . . . . . . . . . . . . . . . . . . . . . . . . . . .

### F.4    USES

**Q: Has the dataset been used for any tasks already?**

A: It is used to benchmark 16 diverse retrieval models for the DORIS-MAE task in Section C.

. . . . . . . . . . . . . . . . . . . . . . . . . . . . . . . . . . . . . . . . . . . . . . . . . . . . . . . . . . . . . . . . . . . . . . . . . . . . . . . . . . . . . . . . . . . . . . .

**Q: Is there a repository that links to any or all papers or systems that use the dataset?**

A: https://github.com/Real-Doris-Mae/Doris-Mae-Dataset

. . . . . . . . . . . . . . . . . . . . . . . . . . . . . . . . . . . . . . . . . . . . . . . . . . . . . . . . . . . . . . . . . . . . . . . . . . . . . . . . . . . . . . . . . . . . . . .

**Q: What (other) tasks could the dataset be used for?**

A: It could serve as a dataset to evaluate query decomposition models.

. . . . . . . . . . . . . . . . . . . . . . . . . . . . . . . . . . . . . . . . . . . . . . . . . . . . . . . . . . . . . . . . . . . . . . . . . . . . . . . . . . . . . . . . . . . . . . .

### F.5    DISTRIBUTION

**Q: Will the dataset be distributed to third parties outside of the entity (e.g., company, institution, organization) on behalf of which the dataset was created?**

A: The dataset is released at Zenodo, and can be accessed https://doi.org/10.5281/zenodo.8035110.

. . . . . . . . . . . . . . . . . . . . . . . . . . . . . . . . . . . . . . . . . . . . . . . . . . . . . . . . . . . . . . . . . . . . . . . . . . . . . . . . . . . . . . . . . . . . . . .

**Q: How will the dataset will be distributed?**

A: The dataset will be distributed on GitHub and Zenodo. Its DOI is https://doi.org/10.5281/zenodo.8035110

. . . . . . . . . . . . . . . . . . . . . . . . . . . . . . . . . . . . . . . . . . . . . . . . . . . . . . . . . . . . . . . . . . . . . . . . . . . . . . . . . . . . . . . . . . . . . . .

**Q: When will the dataset be distributed?**

A: It was released on June 13, 2023.

. . . . . . . . . . . . . . . . . . . . . . . . . . . . . . . . . . . . . . . . . . . . . . . . . . . . . . . . . . . . . . . . . . . . . . . . . . . . . . . . . . . . . . . . . . . . . . .

**Q: Will the dataset be distributed under a copyright or other intellectual property (IP) license, and/or under applicable terms of use (ToU)?**

A: The dataset is distributed under the CC-BY-NC license. For more details, see Section A.1.

### F.6    MAINTENANCE

**Q: Who is supporting/hosting/maintaining the dataset?**

A: The authors on GitHub and Zenodo.

. . . . . . . . . . . . . . . . . . . . . . . . . . . . . . . . . . . . . . . . . . . . . . . . . . . . . . . . . . . . . . . . . . . . . . . . . . . . . . . . . . . . . . . . . . . . . . .

**Q: How can the owner/curator/manager of the dataset be contacted (e.g., email address)?**

A: One could raise issues on GitHub or contact authors by emails, which will be revealed upon paper's acceptance.

....................................................................................

**Q: Will the dataset be updated (e.g., to correct labeling errors, add new instances, delete instances)?**

A: Yes, the dataset will be updated as needed to correct errors.

....................................................................................

**Q: Will older versions of the dataset continue to be supported/hosted/maintained?**

A: DOIs and downloadable links for older versions of the dataset will be documented on GitHub.

....................................................................................

**Q: If others want to extend/augment/build on/contribute to the dataset, is there a mechanism for them to do so?**

A: Yes. For scalability see Section 4.4 and Section B.5. For future work see Section 6.