# OpenReview forum: "Scientific Document Retrieval using Multi-level Aspect-based Queries"
_NeurIPS.cc/2023/Track/Datasets_and_Benchmarks — NeurIPS 2023 Datasets and Benchmarks Poster_

### Official Review · Reviewer_vmkV · 2023-07-18
**officiel review**

**Rating:** 5
**Confidence:** 4
**Clarity:** The writing is good.

**Strengths:**

1. The effort to provide dataset on long queries with multiple aspect is a meaningful direction. There are some interesting perspectives in the paper, such as just use ChatGPT to annotate sub-aspect, which is a simpler task.

2. The paper is clearly written. The annotation process is clear and the process to make sure ChatGPT annotation quality is well discussed, which the reviewer really appreciate.


**Additional Feedback:**

What is the variance of the numbers, such as in Table 2?


**Correctness:**

The annotation of queries only involve 3 graduate students, so the bias can be high.

The construction of the dataset uses ChatGPT for annotation. The authors did strive to prove that ChatGPT annotation is sensible. Still, it odes not align perfectly with humans, and for people using this dataset, ChatGPT annotation does not add extra value than human annotations. So overall the reviewer is more or less convinced about the approach but not fully convinced.

**Documentation:**

The URL is valid. All the files are pickle files so the reviewer does not check closely.

**Limitations:**

No limitation section.

**Opportunities For Improvement:**

On the dataset front:
1. The major concern is dataset scale - there are only 50 queries, in one domain. If one wants to write a paper to study multi-level aspect-based query ranking/retrieval, the paper is not likely to go anywhere by using the dataset introduced in this work.

2. Furthermore, there's no "training data". How can a researcher benefit this significantly if they want to do research, in multi-level aspect-based queries? Just doing evaluation seems like a hit & miss, as what people are doing on BEIR datasets (at least there people get can an average number from multiple datasets). Metrics from 50 queries are likely to be too noisy to be meaningful (if there's only one dataset).

3. The queries are generated by looking at some papers. This is reverse to the natural process, so the reviewer is not sure how meaningful the query distribution is in practice.

On the benchmark front:
1. “trained the models on our CS corpus” - this needs more details. How can one reproduce the results?
2. The benchmark shows that existing methods do not work well, comparing with existing datasets. It is not surprising since the methods are not trained on such queries. This is relevant to a comment above - given this dataset people still can not train on such datasets and there's no clear path to it.

**Relation To Prior Work:**

The difference from [42] needs to be more clear and explicit. So far it looks quite incremental since [42] also studies computer science articles and faceted queries.

**Summary And Contributions:**

The paper proposes a dataset for document retrieval with long queries, where there are multiple aspects. A dataset with 50 queries is constructed. The dataset is only in the computer science paper domain. The queries are manually generated after looking at certain papers.

First, a manual process decomposes each query to several aspects, and each aspect is decomposed to several sub-aspects.

Second, ChatGPT is used to annotate sub-aspect, document pairs. A study is performed to make sure the discrepancy between human annotation and ChatGPT is not large.

Then a benchmark is performed on the dataset for evaluation. The message the authors try to deliver is the numbers look lower than existing datasets.

---

> ### Author Response · Authors · 2023-08-20
> **Author Response (Part 1/2)**
>
> Thank you for your constructive feedback, and for valuing our effort in introducing a dataset for long queries with multiple aspects. We're gratified that our use of LLMs for sub-aspect annotations and the thoroughness of our annotation quality assessment stood out. We now address your specific concerns.
>
> > The major concern is dataset scale - [...] in this work.
>
> Thank you for your feedback on the dataset's scale. Our dataset initially contained 50 complex queries, but also included about 4,000 sub-queries.
>
> In response to your feedback, we have added another 50 complex queries using our annotation framework, bringing the total to 100 (L106-112). As shown in Table 2, this expanded dataset allows for a more precise evaluation of retrieval method performance.
>
> > Furthermore, there's no "training data". [...] in multi-level aspect-based queries?
>
> In response to your comment, we have conducted experiments to demonstrate the utility of our dataset. Specifically, we used the expanded dataset to fine-tune an embedding model and subsequently compared its performance to its pre-trained checkpoint.
>
> Details of our experiments (see the newly added Section 5.3 L328-341 and Table 8):
>
> - Dataset Split: From our dataset, we designated 40 queries for training and 60 for testing.
>
> - Model Fine-Tuning: With the training data, we generated triplets comprising a query and two abstracts. These triplets were used for contrastive fine-tuning of the RoBERTa-based E5-Large-v2 model (355M parameters), over a single epoch without any hyperparameter adjustments.
>
> - Upon evaluation with the 60 test queries, our fine-tuned model consistently outperformed the pre-trained baseline across all metrics. The results, which include mean and standard deviation values, are shown in Table 8 in the revised paper.
>
> > Metrics from 50 queries are likely to be too noisy to be meaningful (if there's only one dataset).
>
> We now report standard deviations for model performance in Table 2 and Table 8. The performance estimates are precise enough to distinguish many models from each other.
>
> > The queries are generated by looking at some papers. [...] query distribution is in practice.
>
> Our dataset represents an initial step towards developing a collection of queries that are both realistic and much more complex than those currently available in existing datasets. While our queries were inspired by papers, our goal was to create questions that embody genuine research concerns, going beyond the simplicity often found in online forums. You can review the queries on our GitHub repository (https://github.com/Real-Doris-Mae/Doris-Mae-Dataset/blob/main/queries.txt) to get a sense of their design. We hope this dataset encourages advancements in models designed to handle such detailed queries.
>
> > 'trained the models on our CS corpus' - this needs more details.
>
> ​​Thank you for your feedback. We trained two models: Specter and SciBERT (see L280).
>
> - For SPECTER, we trained it for one epoch using the implementation provided in its GitHub repository. We used citations in our corpus for supervision.
>
> - For SciBERT, we also trained it for one epoch, starting from the pre-trained checkpoint available at the reference Github repository. For this model, we passed all papers in the corpus for unsupervised training with cross-entropy loss. All hyperparameters were the same as in the original model.
>
> > The benchmark shows that existing methods do not work well, [...] not trained on such queries.
>
> We appreciate your observation. Several models we benchmarked, including SPECTER, ada-002, and SciBERT, have demonstrated strong zero-shot performance on other information retrieval datasets such as MS MARCO. Our evaluations show that these zero-shot capabilities do not extend to DORIS-MAE.

---

> > ### Author Response · Authors · 2023-08-20
> > **Author Response (part 2/2)**
> >
> > This is the second part of our response.
> >
> > > The annotation of queries only involve 3 graduate students, so the bias can be high.
> >
> > We took several steps in order to eliminate bias from the annotations. First, all annotations were performed independently, i.e. without consultation among the annotators. Second, we developed clear annotation guidelines, as detailed in Appendix B1. These guidelines provide a structured framework for the annotators, reducing the potential for bias.
> >
> > The following link contains each query along with the most relevant documents: https://github.com/Real-Doris-Mae/Doris-Mae-Dataset/blob/main/best_candidates.txt
> >
> > We believe that they provide qualitative evidence of the effectiveness of our annotation pipeline. We are happy to provide additional data if it would be useful.
> >
> > > Still, it does not align perfectly with humans, and for people using this dataset, ChatGPT annotation does not add extra value than human annotations.
> >
> > While we acknowledge that ChatGPT's annotation may not align perfectly with human annotations, our findings presented in Table 1 indicate that the difference between ChatGPT and human annotations is not statistically significant. The primary value of ChatGPT's annotation lies in its viability to replace human labor in a cost-effective and efficient manner. We believe this constitutes a valuable contribution to the field.
> >
> > > The difference from [53] (CSFCube) needs to be more clear and explicit. So far it looks quite incremental since [53] (CSFCube) also studies computer science articles and faceted queries.
> >
> > We would like to highlight the following key differences:
> >
> > - Annotation Method: Unlike the CSFCube paper [53] which relies on human annotations, our study employs an automatic annotation tool. This allows us to scale up to more queries and larger candidate pools efficiently.
> >
> > - Aspect Design: The CSFCube paper [53] manually designs three aspects: “background”, “methodology”, and “results”. In contrast, our approach uses open-ended textual descriptions for aspects/sub-aspects. For specific examples, please refer to https://github.com/Real-Doris-Mae/Doris-Mae-Dataset/blob/main/queries.txt
> >
> > - Research Focus: While the CSFCube paper [53] concentrates on calculating the similarity between two abstracts, our research is centered on the relevance between complex queries and abstracts. This is a promising and relatively unexplored research direction.
> >
> > > What is the variance of the numbers, such as in Table 2
> >
> > We now provide standard deviations for each metric in Table 2 in the revised version of the main paper.

---

> ### Author Response · Authors · 2023-08-27
>
> Thank you again for your helpful review of our paper. We wanted to check whether our responses have addressed your questions, or whether there is additional information that would be useful.

---

> > ### Comment · Reviewer_vmkV · 2023-08-29
> >
> > Thanks for the replies. Will take the comments into account during review discussions.

---

### Official Review · Reviewer_xrVv · 2023-07-20
**Exciting new dataset in complex multifaceted scientific search, but limited to ML-flavoured papers**

**Rating:** 7
**Confidence:** 4

**Strengths:**

- Complex, multifaceted queries for scientific document retrieval is an important and exciting area of research.
- There is a lack of labelled data and benchmarks for this task, so the DORIS-MAE dataset fills an obvious need and benefits the IR research community.

**Additional Feedback:**

N/A

**Clarity:**

- On lines 24-25, when it is mentioned that "Scientists often engage in complex inquiries...", it would have been nice to see an example or even a cluster of examples inline, to give a flavour of the type of complex queries you have in mind. If space is a concern, Figure 1 could be repurposed for this, e.g. by citing it on line 25. Ditto for the point about multifaceted queries in the next few lines --- I know what you mean, but examples here would have been great.
- I think the "motivations" section, which appears to start on line 40, could use some work. The first paragraph is easy to understand, but the introduction of Anno-GPT comes out of nowhere on line 47. The subsequent paragraph feels more related to the first one? And there is redundancy at the end as the main contributions are listed again. My recommendation would be to: 1) first introduce the dataset and explain its motivation, 2) explain Anno-GPT, and finally, 3) explain your main results and conclusions. Remove or merge the redundant content in lines 60-65.

__Typos, grammatical errors, or awkward phrasing__

- The first sub-aspect of Figure 1 might have a typo: "[...] describe a model that the instruct the action [...]"
- What does NIR stand for? (is it "neural" IR?) It's first given as an acronym on line 56, but I don't think it's ever given in its unabbreviated form.

**Correctness:**

- On lines 33-34, it is mentioned that current IR systems and search engines fall short when dealing with complex, multifaceted queries. I agree, but it would have strengthened the point if you could cite work in this area with evidence of some shortcomings of modern IR systems. One such work that I can think of is https://aclanthology.org/2022.tacl-1.13/. The same critique applies to lines 72-74 and 79-80.
- I am struggling a bit to convince myself of the soundness of the approach used to create the candidate pool. The first two sections of the paper spend a lot of time claiming that existing IR methods are not up-to-snuff for multifaceted queries, _only to use them to collect the ground-truth set of documents for each query_. I am guessing the thinking here is that the IR methods were given _single_-aspect queries (if so, fair enough). My main concern is that, even using the citation graph, there will be __false negatives__ in the candidate pool (hopefully __false positives__ should get low relevance scores during annotation, Section 4.1). Do the authors examine any of the candidate pools to get a sense of their recall? Finally, the authors go so far as to call this a "ground-truth" in Section 3 (lines 122), but I think it's closer to a silver standard.
- On lines 75-77, it is said: "Other models such as SPECTER [...] focus on calculating document-level similarity. However, these models are largely designed to work starting from an existing query paper, which makes them less suited for open-ended queries". This was true of the original SPECTER, but it seems V2 now accepts open-ended queries. Given that, I would amend this statement and maybe consider including V2 as another retriever baseline (see __Relation To Prior Work__).

**Documentation:**

The dataset is extensively documented in the supplementary material, which includes a maintenance plan. The dataset is available online, freely, via Zenodo.

**Limitations:**

A clear limitation of this work is that it is restricted to ML-flavoured scientific papers. I would be willing to bet the style of query for complex scientific questions varies quite a bit across fields. As it stands, it is sort of glossed over by mentioning that Anno-GPT _could hypothetically_ be used to annotate examples from other domains.

However, I would have expected more of a discussion on this limitation in the paper and its possible consequences, e.g. would a system trained or developed on DORIS-MAE be reasonably expected to extend to other domains? Or would an entirely new dataset need to be produced? What would be the benefits or trade-offs of a _mixed domain_, multi-level aspect-based query annotated dataset?

**Opportunities For Improvement:**

- The dataset is rather small (50 queries) and limited to ML-flavoured CS papers, which are not representative of the scientific literature at large (this point is fleshed out below under __Limitations__). However, the authors note that from the original 50 queries, 4000 subqueries can be crafted and highlight a workflow for automatically labelling more data.
- It is unclear to me if the method used to create the candidate pool of documents per query is appropriate. I flesh out my thinking below under __Correctness__.
- I think the framing of "Anno-GPT" is creeping into over-claiming territory. It is a neat approach, and I commend the authors' thoroughness. However, automatically labelling data using LLMs like ChatGPT has become almost commonplace. The other aspects of the Anno-GPT framework, like tuning the prompt on a dev set, are certainly not novel. This is all fine, but its framing (e.g. "we [...] introduce Anno-GPT, a scalable framework for evaluating the viability of Large Language Models (LLMs) [...] for expert-level dataset annotation tasks") and the amount of space it takes up in the paper both seem a bit exaggerated to me when compared to the actual contribution.

**Relation To Prior Work:**

- There is a newer version of SPECTER available that might have been a stronger baseline in your experiments. Code: https://huggingface.co/allenai/specter2. Unclear if SPECTER V1 or V2 was used but I am assuming the former.

**Summary And Contributions:**

This paper studies the problem of retrieving relevant scientific abstracts given a complex, multifaceted query. The major contributions include:

- The release of a new dataset (__DORIS-MAE__) consisting of 50 human-authored, multifaceted queries, each with ~100 relevant documents scored with relevance judgments, from which over 4000 subqueries can be crafted.
- A careful and thoughtful approach using ChatGPT to produce the relevance scores automatically.
- An analysis of many existing retrievers on the newly introduced DORIS-MAE dataset.

---

> ### Author Response · Authors · 2023-08-20
> **Author Response (Part 1/2)**
>
> Thank you for your valuable feedback. Your acknowledgment of the importance of multifaceted queries and the dataset's role in filling a research gap is greatly appreciated. We now address your specific concerns.
>
> > The dataset is rather small (50 queries).
>
> We understand your concern about the size of DORIS-MAE. To illustrate the scalability of the annotation framework, we have employed it to annotate an additional 50 complex queries along with their corresponding candidate pools in the same manner as before. This has doubled our total dataset size to 100 complex queries, which is sufficient to estimate model performance with reasonable precision (see L106-112).
>
> > A clear limitation of this work is that it is restricted to ML-flavoured scientific papers. [...] annotate examples from other domains.
>
> We have updated the paper (L357-360) to note this limitation. We have also clarified our claim that Anno-GPT could be used to annotate other domains. We make two primary contributions in this area. First, explicitly identifying the risk of overfitting when developing an LLM-based annotation pipeline. This is somewhat counterintuitive, since no models are being trained. Second, providing a procedure, shown in L199-208, which can be used to develop and validate language model annotators in a statistically sound manner.
>
> > I am struggling [...] to create the candidate pool. [...] there will be false negatives in the candidate pool [...] Do the authors examine any of the candidate pools to get a sense of their recall?
>
> Thank you for your feedback. We recognize the potential for false negatives in our dataset and appreciate the importance of this issue. Given that there are around 20k abstracts in each category of our dataset (AI, ML, CV, NLP), fully quantifying the number of false negatives would require annotating all of these abstracts for every query. This would be a large undertaking, with an estimated cost of $10,000.
>
> We believe that the primary risk from false negatives is that the downstream reranking task will be impossible, due to the absence of relevant documents. We take a two-pronged approach to evaluating this risk.
>
> Qualitative Evaluation of Relevance: In our supplementary materials, we have now highlighted the most relevant abstract identified for each query (available at https://github.com/Real-Doris-Mae/Doris-Mae-Dataset/blob/main/best_candidates.txt). A qualitative review suggests that the abstracts are generally relevant to the query.
>
> Comparative Analysis with Alternative Retrieval Methods: Given that the original candidate pool was constructed primarily from keyword and citation signals (accounting for ~85% of the sources), we aimed to determine if independent retrieval methods could yield better results. We assessed two embedding methods, E5-Large-v2 and SPECTER-v2, to create what we refer to as the 'embedding pool' for each query. This new pool comprised 150 abstracts. Our goal was to compare the most relevant document from both the original and embedding pools.
>
> The results indicate that, on average, the most relevant documents in the original pool scored higher than those in the embedding pool by 4%. However, for 30% of the queries, the embedding pool did contain a more relevant document, with an average improvement of 12% over the original pool.
>
> This analysis suggests that while there's potential for improvement, the top documents in our original pool are approximately as good as those discovered through independent retrieval methods.
>
> > Finally, the authors go so far as to call this a "ground-truth" in Section 3 (lines 122), but I think it's closer to a silver standard.
>
> Thank you for pointing this out. We changed it to a more conservative and accurate description in L123.
>
> > I think the framing of "Anno-GPT" is creeping into over-claiming territory. [...] exaggerated to me when compared to the actual contribution.
>
> Thank you for raising this concern. One of our contributions here is recognizing the entire annotation pipeline – encompassing query decomposition strategy, scoring criteria, and prompt selection – as a process vulnerable to overfitting. Additionally, we introduce a straightforward procedure to address this issue.We have recalibrated the paper’s tone (L188-198), and provide a clearer description of our work.

---

> > ### Author Response · Authors · 2023-08-20
> > **Author Response (Part 2/2)**
> >
> > This is the second part of our response.
> >
> > > However, I would have expected more of a discussion [...] system trained or developed on DORIS-MAE [...] mixed domain [...] annotated dataset?
> >
> > Though DORIS-MAE is initially designed as an evaluation benchmark to probe the ability of Information Retrieval (IR) models to comprehend deeper and multifaceted semantic components in queries and passages, we believe your suggestion would strengthen our paper. We will include a short discussion of it in our revised paper (Section 6 L357-360).
> >
> > We are uncertain if a model fine-tuned on CS DORIS-MAE would transfer its performance to another field. This is difficult to currently evaluate because of an absence of other datasets with complex query structure. Ideally, a fine-tuned model would learn transferable semantic features, irrespective of the CS domain. However, it is possible a model could instead pick up only CS-specific patterns or knowledge. For scientific search engines for a specific field, the most conservative recommendation would be to create a domain-specific dataset following our annotation procedure.
> >
> > > On lines 33-34, it is mentioned that current IR systems and search engines fall short [...]One such work [...] The same critique applies to lines 72-74 and 79-80.
> >
> > Thank you for this suggestion. We have updated our discussion with this reference (L36-38 & L76-77).
> >
> > > This was true of the original SPECTER, [...] V2 as another retriever baseline
> >
> > Thank you for your suggestion. We have added SPECTER v2 as a baseline in Table 2.
> >
> > > On lines 24-25, when it is mentioned that "Scientists often engage in complex inquiries...", [...] but examples here would have been great.
> >
> > Thank you for this suggestion, we now refer to Figure 1 in L25.
> >
> > A text file detailing the queries and their respective aspects is at (https://github.com/Real-Doris-Mae/Doris-Mae-Dataset/blob/main/queries.txt).
> >
> > > I think the "motivations" section, [...] My recommendation [...] redundant content in lines 60-65.
> >
> > Thank you for your suggestion! We updated the text with these changes. See Section 1 (L49-57).
> >
> > > The first sub-aspect of Figure 1 might have a typo: "[...] describe a model that the instruct the action [...]
> >
> > Apologies for the typo, it should read “[...] describe a model that can instruct the action [...]”, we have updated Figure 1 in the revised paper.
> >
> > > What does NIR stand for? (is it "neural" IR?) [...] its unabbreviated form.
> >
> > Thank you for raising this issue. We now explicitly introduce it as an abbreviation for Neural IR. See L34.

---

> ### Author Response · Authors · 2023-08-27
>
> Thank you again for your helpful review. Since the review period will end on Aug. 29, we wanted to check whether our responses addressed the questions in your review. We are happy to provide additional information if useful.

---

> > ### Comment · Reviewer_xrVv · 2023-08-28
> > **Authors comprehensively addressed my concerns**
> >
> > The authors comprehensively addressed all my concerns, and it looks like the did the same for the other reviewers. In light of this, I am happy to raise my score.

---

### Official Review · Reviewer_uQNo · 2023-07-20
**Dataset for hierarchical aspect based queries constructed with novel annotation framework deploy ChatGPT**

**Rating:** 8
**Confidence:** 4

**Strengths:**

The major contribution is the introduction of a dataset of human-authored query case which are long and are composed of multiple aspects.
This is an imporant challange as in specialized domains humans do requires long and multi-faceted questions.
This can trigger furhter development of novel approaches in IR.
The hierarchical aspect-based structure for instance allows to investigate models to automaically break down queries into aspects
and multi-turn question answering systems.

The second relevant contribution is the AnnoGPT. The authors detailad a systematic and scientifically sound approach to deploy LLM such as ChatGPT to annotate datasets with relevance labels, a task which is notably laborious and time consuming.


**Additional Feedback:**

Maybe it's just me but I find the "Scientific Document Retrieval" in the title a bit misleading.
As queries and documents are exclusively from the CS domain I would explicitly say it in the title.
"Scientific documents" could mean anything from "biology" to "physics".


**Clarity:**

The paper is well written: I coudl follow all arguments.

Two very very minor remarks:

- Abstract: "complex" is repeated

- "Decomposing Queries to Aspects": when listing  it would be visually helpful to have "criteria are: (i)..., (ii) ..., (iii), ..."


**Correctness:**

The dataset is constructed in a scientifically rigorous way and the authors provide a repository with all the code necessary to reproduce the banchmarking results.

**Documentation:**

Code to reproduce the benchmark is provided.
Maintenance plan is not explicitly reported.

**Ethics:**

The dataset does not warrant further discussion on ethical considerations.


**Limitations:**

I haven't found an explicit mention of self-reported limitations.
My suggestions in "Opportunities for Improvement", if not addressable directly, can be listed as such.

**Opportunities For Improvement:**

In "3.3 Candidate Pooling" the authors state that to create the candidate pool to be annotated with relevance scores
they use both standard IR techniques such as BM25 and a OpenAI embedding model.
Each query has 100 documents associated to it, and 90% of the pool is sourced from the first type of models.
This introduce a strong "selection bias" towards lexical matching algorithms (see Section 6 in https://openreview.net/forum?id=wCu6T5xFjeJ). As this is a well documented issue, in my opinion the authors should
motivate this decision and provide a more detailed discussion of the results.

Secondly, as these are long, complex and multi-aspect queries I was expecting to see (even only a subset) of the dataset containing full text articles. The hierarchical structure has a perfect use-case in full text document retrieval, as in this scenario multiple aspects might be discussed in different sections of the articles.
With this being said, I am well awere of the difficulties of annotating full text articles so this is not an issue per se but a suggestion
for future releases if planned.


**Relation To Prior Work:**

The authors describe all the relevant studies in the field.

**Summary And Contributions:**

The authors introduce a novel dataset comprising 50
each associated  with 100 ranked documents.
The major contribution of the dataset is that each query is hierarchically strucutred according to **semantically distinct** aspects.
The second major contribution of the apper is the annotation framework called AnnoGPT,
which can use LLM (ChatGPT in this case) to significantly reduce human effort in labeling documents with relevance scores.
Finally the authors perorm an extensive benchmarking of existig IR and neural IR approaches on the dataset,
and find that it is significantly more challenging than traditional ones.

---

> ### Author Response · Authors · 2023-08-21
> **Author Response**
>
> We thank you for your review, and for highlighting the potential of our dataset and annotation approach. Your acknowledgment of their utility for future IR developments is encouraging. We now turn to your specific comments.
>
> > In "3.3 Candidate Pooling" the authors state that to create the candidate pool [...] "selection bias" [...] provide a more detailed discussion of the results.
>
> Thank you for pointing out the potential selection bias in our candidate pool creation. In response, we generated a new 'embedding pool' using E5-v2 and SPECTER-v2, selecting 150 abstracts for each query. We conducted two main analyses:
>
> - Relevance Analysis: First, we evaluated whether the new pool contains documents which are as relevant as those in the original pool. On average, the most relevant documents in the original pool scored higher than those in the embedding pool by 4%. However, for 30% of the queries, the embedding pool did contain a more relevant document, with an average improvement of 12% over the original pool. This indicates that the relevance levels of both pools are comparable. We are happy to provide more analyses if needed.
>
> - Reranking Performance Analysis: Evaluating reranking on both pools showed a significant drop in performance for the embedding pool. Specifically, Recall@5 decreased by over 15% for most models, except for ada-002 which had similar results across the two pools. We believe this drop might be due to the increased similarity between the models used for pool creation and reranking, making it harder for the reranking model to differentiate the pre-selected documents.
>
> Please let us know if these analyses address your concerns.
>
> > Secondly, as these are long, complex [...] full text articles. [...] but a suggestion for future releases if planned.
>
> Thank you for suggesting the use of decomposition strategies on the full texts of scientific articles. We believe that this is a promising direction for future work.

---

> > ### Comment · Reviewer_uQNo · 2023-08-25
> >
> > Thank you for addressing my comments. I would suggest to include these two analyses in an additional appendix reporting and reference it in the "Candidate Pooling" section.

---

> > > ### Author Response · Authors · 2023-08-27
> > >
> > > Thank you again for your suggestions. We will add these analyses to the revised paper.

---

### Official Review · Reviewer_gbm8 · 2023-08-02
**Interesting resource and annotation experiment; scattered presentation.**

**Rating:** 7
**Confidence:** 5
**Clarity:** In its current form the paper is quit…

**Strengths:**

- The paper presents a useful resource for an interesting task.
- The dataset seems well documented.
- The annotation experiment in the paper is reasonably conducted and could be illustrative for future work.
- The experiments in the paper seem thorough.

**Additional Feedback:**

Provided above.

**Correctness:**

While the complete details of construction are missing or scattered the dataset seems to be constructed in a reasonably sound manner.

**Documentation:**

Sufficient.

**Ethics:**

None.

**Limitations:**

Not discussed.

**Opportunities For Improvement:**

- The paper content is disorganized and information about important elements of the paper seem to be hard to find. Please clarify the following:
	- Sec 3: I understand that there are 50 complex queries. What were the instructions given to the annotators for authoring the complex queries? It seems queries were authored naturally at first and then broken down into aspects/sub-aspects? How were the aspects and sub-aspects identified by the annotators? How many aspects are there per complex query? Is the complex query simply a concatenation of all the aspects/sub-aspects?
	- L117-120 and L154-156: How is a sub query constructed? Is this done automatically? Is any arbitrary set of sub aspects likely to be a realistic/well-formed query?
	- Sec 3.1: Is the paper used for authoring the query a part of the candidate set of documents annotated for relevance?
	- Sec 4: What was the input shown to a user or a model when having them make a relevance judgement? Was it a complete complex query or sub-aspect/aspect? The supplementary section (B.2) attached seem to indicate that the LLMs were only shown individual aspects.
	- Sec 5: How exactly was the model denoted with ID trained? Please describe the precise model, the data, and training details (eg. loss function, negative sampling, optimizer etc)
	- L280-282: Please consider rephrasing the claims about the Aspire model. To the best of my knowledge, the Aspire models don't represent aspects explicitly. They only do so through contextualized sentence embeddings, and obtain aggregated document similarities by aggregating sentence similarities - so while the evaluation on CSFCube evaluated the method on fixed aspects the model is capable of representing any general set of aspects.
	- Sec 5: Were the automatic relevance judgements from ChatGPT used for evaluation?
	- Sec 5.1: I appreciate the numerous experimental comparisons but as currently presented none of the details of the experiments are apparent in the paper. Please consider moving majority of the experimental result tables (Tables 3-7 seem useful but not crucially important) to an appendix and describe in greater detail the baselines attempted and discuss the results of the observed results in much greater depth. For example, it is surprising that an approach which is intended for short queries (RocketQA) in a different domain ends up outperforming several in-domain and more specialized models.
	- Sec 5: It is unclear from the paper precisely what the input to a retrieval model is, it sees like diversity of types of models were evaluated eg. short query retrieval models (like colbert or rocketqa), paragraph level models (like specter), aspect conditional paragraphs (like Aspire) - what exactly was the input to each of these models? Please clarify this with an example. I see that there is some discussion of this in the appendix, please consider moving this to the main paper body.
	- Related to the above point, I would highly encourage addition of a clear task description section - please describe the various tasks possible with your dataset and their corresponding inputs and outputs - please make an attempt to describe the task independently of downstream models which would be applicable to the task. This is specially important since this paper seems to introduce a non-trivially different retrieval task.

**Relation To Prior Work:**

Please consider discussing work on the automatic relevance estimation in IR (https://arxiv.org/abs/2304.09161, https://arxiv.org/abs/2302.11266) and narrative-driven recommendations (https://dl.acm.org/doi/10.1145/3109859.3109893, https://arxiv.org/abs/2105.09204, https://arxiv.org/abs/2306.02250) - They are closely related to the annotation approach and task presented.

**Summary And Contributions:**

The paper introduces a test collection for an information retrieval task with complex multi aspect narrative queries. Further the paper presents an experiment which uses ChatGPT and demonstrates its feasibility for annotating query-document pairs in the dataset. Finally the paper reports on several standard and specialized baselines for the proposed task.

---

> ### Author Response · Authors · 2023-08-20
> **Authors' Response**
>
> Thank you for your valuable feedback. We appreciate your recognition of our dataset’s value, the thoroughness of our experiments, and the potential of our annotation method as a model for future work. We now turn our attention to your specific comments and concerns.
>
> > I understand that there are 50 complex queries. [...] Is the complex query simply a concatenation of all the aspects/sub-aspects?
>
> The guidelines for query formation are now described in L134-144, and the guidelines for query decomposition are described in L146-166. The complex query is not a concatenation of the aspects. As outlined in L111-112, a single query can have up to 9 aspects, and each aspect can have up to 6 sub-aspects. A text file containing the queries and their aspects is at (https://github.com/Real-Doris-Mae/Doris-Mae-Dataset/blob/main/queries.txt).
>
> >  How is a sub query constructed? Is this done automatically? Is any arbitrary set of sub aspects likely to be a realistic/well-formed query?
>
> A subquery is constructed by: select a subset of the aspects, find the corresponding sentences in the query, and then concatenate those sentences. The process is described in: L124-125, and is an automatic process. Qualitatively, the sub queries are well-formed. They are available at the following link: (https://github.com/Real-Doris-Mae/Doris-Mae-Dataset/blob/main/subquery-2.txt)
>
> The subqueries are realistic queries with lower complexities (see L324-325 for a discussion). Models’ performances on the subqueries are discussed in Table 7 and L320-324.
>
>  > Sec 3.1: Is the paper used for authoring the query a part of the candidate set of documents annotated for relevance?
>
> The paper(s) used for authoring the query is **not** a part of the query’s candidate pool.
>
> > Sec 4: What was the input shown to a user or a model [...] LLMs were only shown individual aspects.
>
> Both annotators and models receive identical inputs – individual aspects paired with a candidate abstract from the pool.
>
> > Sec 5: How exactly was the model denoted with ID trained? [...] optimizer etc)
>
> The retraining details are at Section 5 (L280).
>
> For SPECTER, we trained it for one epoch using the implementation provided in the GitHub repository. We utilized the citation signal present in the corpus as supervising signals and employed the provided training code for in-domain training. Here is the link to the GitHub repository for your reference: https://github.com/allenai/specter.
>
> For SciBERT, we also trained it for one epoch. We used the pre-trained checkpoint available at https://github.com/allenai/scibert/. For this model, we passed all papers in the corpus for unsupervised training with cross-entropy loss. All hyper-parameters were the same as in the original model.
>
> Please let us know if there is additional information that would be useful.
>
> > L280-282: Please consider rephrasing the claims about the Aspire model. [...] general set of aspects.
>
> Thank you for the suggestion. We rephrased our discussion of ASPIRE in L289-291 & 296-297.
>
> > Sec 5: Were the automatic relevance judgements from ChatGPT used for evaluation?
>
> Yes, the score we get from ChatGPT’s annotation is aggregated to provide rankings for the candidate pool which is used for model evaluation, as mentioned in our revised paper (L119-122 and Equation 1).
>
> > Sec 5.1: I appreciate the numerous experimental comparisons [...] (RocketQA) [...] in-domain and more specialized models.
>
> For models that are intended for short queries (Rocket QA and Sent-BERT), we processed the input in two ways: 1) each sentence of the query is processed independently, and the averaged cosine similarity is used to rank the candidates, as discussed in L286-289; and 2) the input query is provided to the model as a single text string. To determine the overall model performance, we took the maximum of the performance achieved by 1) and 2). We also removed 2 tables from the original submission’s Tables 3-7, per your suggestion.
>
> > Sec 5: It is unclear from the paper precisely what the input to a retrieval model is, [...] what exactly was the input to each of these models?
>
> The input is either sentence-by-sentence or the entire query. No aspect information is given. We chose the input format appropriate for the specific model. Evaluation details are given in Section C in the Appendix.
>
> > Related to the above point, I would highly encourage addition of a clear task description section [...] introduce a non-trivially different retrieval task.
>
> We now give our task description in L42-48. The task of DORIS-MAE is: given a complex multi-aspect Q, and a set of candidate pools, rank the candidates based on their relevance to Q, and retrieve the relevant candidates.
>
> > Please consider discussing work on the automatic relevance estimation [...] They are closely related to the annotation approach and task presented.
>
> Thank you for the suggestions, we now discuss these papers in the revised version (L94-96, 101-102).

---

> > ### Comment · Reviewer_gbm8 · 2023-08-24
> > **Thank you for the clarifications.**
> >
> > Thank you for the clarifications. I have raised my score.

---

### Official Review · Reviewer_XCsh · 2023-08-04
**Complex query document retrieval dataset and LLM based annotation framework**

**Rating:** 9
**Confidence:** 4
**Clarity:** Yes the paper has sufficient level of…

**Strengths:**


The idea and process behind generating sub-queries or aspects of a complex query seems to be sound. The evaluation of chatGPT generated annotations and comparison with human annotations is thorough.

The method of auto-generating annotations for complex queries leads to a significant reduction in costs and time (1000 times and 500 times respectively). This would be useful for the broader research community in

I like the fact that the complexity of the dataset is compared with other datasets for several models.

**Additional Feedback:**

The limitations discussion is a bit light, its more about the future work possibilities but there could be more discussions on how the existing methods (including both query decomposition and anno-GPT)  have their limitations in achieving higher metric scores.

**Correctness:**

Yes, the claims in the submission are correct, dataset is constructed in a sound way and the benchmarking of the dataset is done with correctly chosen models,  experiments are executed soundly. All of the standard Information Retrieval evaluation metrics are chosen.

**Documentation:**

Sufficient documentation for the dataset exists at https://github.com/Real-Doris-Mae/Doris-Mae-Dataset. The supplementary material  thoroughly answers all questions regarding documentation.

**Ethics:**

I don't see any ethical feedback considerations as the dataset consists of queries and documents regarding scientific literature.

**Limitations:**

As mentioned above, the limitations section is a bit light, its more about the future work possibilities but there could be more discussions on how the existing methods (including both query decomposition and anno-GPT) have their limitations in achieving higher metric scores.

**Opportunities For Improvement:**

The limitations section is a bit light, its more about the future work possibilities but there could be more discussions on how the existing methods (including both query decomposition and anno-GPT)  have their limitations in achieving higher metric scores.

There could be more queries used, at the moment 50 queries appear a small set.

**Relation To Prior Work:**

Yes related work section is exhaustive, considering that some of the topics covered in the paper are relatively novel.

**Summary And Contributions:**

This paper proposes a new dataset (DORIS-MAE) comprising 50 unique complex queries in the computer science (CS) domain, paired with ranked pools of relevant CS article abstracts. Each query is organised into a hierarchical structure of aspects/sub-aspects to help annotation. They also release a large language model based pipeline called Anno-GPT which generates annotations. The quality of these annotations are shown to be comparable to human annotators. The speed of these LLM generated annotations far outpace human annotators (12 hours vs 5573 hours).

The paper then benchmarks the dataset against a number of state-of-the-art models.

Lastly, the paper also compares performance of these models on this DORIS-MAE dataset vs other standard public datasets like MS-MARCO. The state-of-art models have significantly lower performance on the DORIS-MAE dataset, confirming its high level of complexity of queries.

---

> ### Author Response · Authors · 2023-08-21
> **Author Response**
>
> Thank you for your feedback. We appreciate your acknowledgement of the thoroughness of our ChatGPT evaluation, the utility of our auto-generation process, and the benchmarking with other datasets. We address your specific suggestions and concerns in the following sections.
>
> > The limitations section is a bit light, [...] in achieving higher metric scores.
>
> Thank you for this suggestion. We have extended our limitation section, and added a discussion of this issue. (See Line 357-365)
>
> > There could be more queries used, at the moment 50 queries appear a small set.
>
> Thank you for highlighting the dataset size. In response, we have expanded DORIS-MAE to demonstrate the scalability of our annotation framework. We have added another 50 complex queries and annotated their corresponding candidate pools, doubling the size of the dataset to 100 queries. Please refer to lines L106-107 for details. The results in Table 2 have been updated to reflect this expansion.

---

> > ### Comment · Reviewer_XCsh · 2023-08-29
> >
> > Yes, I see that the suggestions I made have been addressed. Well done to the authors for adding more queries and reflecting more on the limitations of their approach.

---

### Decision · Program_Chairs · 2023-09-22

**Decision:**

Accept (Poster)

**Comment:**

This paper presents a new dataset (DORIS-MAE) consisting of 50 complex computer science queries, each paired with 100 ranked relevant document abstracts. The queries are organized hierarchically into semantic aspects, facilitating annotation. The authors introduce AnnoGPT, an annotation framework utilizing ChatGPT to expedite document labeling, demonstrating its efficiency compared to human annotators. The paper benchmarks state-of-the-art models on the DORIS-MAE dataset, revealing its high query complexity. Furthermore, it compares these models' performance on DORIS-MAE to standard public datasets like MS-MARCO, confirming the dataset's increased difficulty.  The reviewers have expressed a favorable view of the work. It is evident that the authors have dedicated significant efforts towards addressing the reviewers' comments and concerns. In light of the diligence exhibited by the reviewers in their evaluation, it is advisable for the authors to meticulously consider and incorporate all pertinent feedback into their paper prior to the final submission.